# Stochastic cell-cycle entry and cell-state-dependent fate outputs of injury-reactivated tectal radial glia in zebrafish

**Shuguang Yu[1,2], Jie He[1]\***

[1]State Key Laboratory of Neuroscience, Institute of Neuroscience, Shanghai Institutes for Biological Sciences, Center for Excellence in Brain Science and Intelligence Technology, Chinese Academy of Sciences, Shanghai, China; [2]University of Chinese Academy of Sciences, Beijing, China

**Abstract** Gliosis defined as reactive changes of resident glia is the primary response of the central nervous system (CNS) to trauma. The proliferation and fate controls of injury-reactivated glia are essential but remain largely unexplored. In zebrafish optic tectum, we found that stab injury drove a subset of radial glia (RG) into the cell cycle, and surprisingly, proliferative RG responding to sequential injuries of the same site were distinct but overlapping, which was in agreement with stochastic cell-cycle entry. Single-cell RNA sequencing analysis and functional assays further revealed the involvement of Notch/Delta lateral inhibition in this stochastic cell-cycle entry. Furthermore, the long-term clonal analysis showed that proliferative RG were largely gliogenic. Notch inhibition of reactive RG, not dormant and proliferative RG, resulted in an increased production of neurons, which were short-lived. Our findings gain new insights into the proliferation and fate controls of injury-reactivated CNS glia in zebrafish.
DOI: https://doi.org/10.7554/eLife.48660.001

**\*For correspondence:**
jiehe@ion.ac.cn

**Competing interests:** The authors declare that no competing interests exist.

## Introduction

Traumatic brain injury (TBI) is one clinically principal type of central nervous system insults (*Burda and Sofroniew, 2014*). Gliosis defined as reactive changes of resident macroglia (*e.g.*, mammalian astrocytes) is a primary CNS response to TBI in mammals (*Barres, 2008*; *Burda and Sofroniew, 2014*). In mammals, gliosis undergoes three significant stages: Glial cells initially become reactive, hypertrophic, and inflammatory, with characteristic upregulation of GFAP and vimentin (*Liddelow and Barres, 2017*; *Zamanian et al., 2012*); subsequently, a subset of reactive glia re-enter the cell cycle and become proliferative (*Gallo and Deneen, 2014*); finally, proliferative glia undergo gliogenesis, the process of glial cell production, and form structures known as glial scars (*Burda and Sofroniew, 2014*). Earlier studies have demonstrated both protective and detrimental roles of the gliosis in the injured CNS (*Faulkner et al., 2004*; *Li et al., 2008*; *Silver and Miller, 2004*; *Sofroniew and Vinters, 2010*; *Wanner et al., 2013*). For instance, the blockage of initial glia reactivation worsened the injury (*Faulkner et al., 2004*; *Li et al., 2008*; *Wanner et al., 2013*), whereas glia scars hindered neuronal regeneration (*Silver and Miller, 2004*). In the process, the proliferation and fate controls of injury-reactivated RG are essential but remain elusive in vivo.

In contrast to the mammalian CNS, teleost fish exhibit a superior neural regeneration in response to TBI beyond embryonic development (*Baumgart et al., 2012*; *Grandel et al., 2006*; *Kishimoto et al., 2012*; *Reimer et al., 2008*; *Than-Trong and Bally-Cuif, 2015*). Radial glia (RG), the primary form of macroglia in teleost fish, are the main cell source for injury-mediated regeneration (*Than-Trong and Bally-Cuif, 2015*). For instance, RG of different brain regions in zebrafish, including the telencephalon, the hypothalamus, and the spinal cord can produce newborn neurons

**eLife digest** The brain contains networks of cells known as neurons that rapidly relay information from one place to another. Other brain cells called glial cells perform several roles to support and protect the neurons including holding them in position and supplying them with oxygen and other nutrients.

Damage to the brain as a result of physical injuries is one of the leading causes of death and disability in people worldwide. Brain injuries generally stimulate glial cells to enter a "reactive" state to help repair the damage. However, some glial cells may start to divide and produce more glial cells instead, leading to scar-like structures in the brain that hinder the repair process.

To investigate why brain injuries trigger some glial cells to divide, Yu and He systematically examined glial cells in the part of the zebrafish brain that handles vision, known as the optic tectum. The experiments showed that a physical injury stimulated some of the glial cells to divide. Repeated injuries to the same part of the brain did not always stimulate the same glial cells to divide, suggesting that this process happens in random cells.

Further experiments revealed that molecules involved in a signaling pathway known as Notch signaling were released from some brain cells and inhibited neighboring glial cells from dividing to make new glial cells. Unexpectedly, inhibiting Notch signaling after a brain injury caused some of the glial cells that were in the reactive state to divide to produce neurons instead of glial cells.

Understanding how the brain responds to injury may help researchers develop new therapies that may benefit human patients in future. The next steps following on from this work will be to find out whether glial cells in humans and other mammals work in the same way as glial cells in zebrafish.
DOI: https://doi.org/10.7554/eLife.48660.002

in response to the injury (*Duncan et al., 2016*; *Goldshmit et al., 2012*; *Johnson et al., 2016*; *Kizil et al., 2012*; *Kroehne et al., 2011*; *Than-Trong and Bally-Cuif, 2015*). Also, retinal müller glia (MG) can be reactivated by the injury, giving rise to newborn retinal neurons (*Goldman, 2014*; *Gorsuch and Hyde, 2014*).

Molecular mechanisms underlying the proliferation and fate controls of injury-reactivated RG have been examined in zebrafish for many years (*Dias et al., 2012*; *Goldman, 2014*). As to the proliferation control, Notch signaling is involved but is somehow context-dependent. For instance, in the zebrafish spinal cord, glial cells have low levels of Notch activity when they are in the dormant state, and enter the cell cycle by increased Notch activity after the injury (*Dias et al., 2012*). In contrast, dormant RG of zebrafish telencephalon exhibit high Notch activity and become proliferative by a rapid decrease of Notch activity (*Chapouton et al., 2010*). Notch signaling has also been reported to regulate fate outputs of reactivated MG in the injured zebrafish retina, that is, Notch inhibition leads to gliogenesis, whereas Notch over-activation results in the production of photoreceptor cells (*Wan et al., 2012*).

Zebrafish optic tectum, the higher sensory integration center, possesses a large population of RG (*Galant et al., 2016*; *Ito et al., 2010*). Unlike other brain regions where RG present as both dormant and proliferative forms at physiological conditions, tectal RG has been reported to be dormant, and are reactivated by injury to give rise to newborn neurons via Wnt signaling as well as Notch signaling (*Shimizu et al., 2018*; *Ueda et al., 2018*). Interestingly, a recent study showed that tectal RG produced a significant number of glial cells (~25%) but not neurons (*Lindsey et al., 2019*). It is essential to resolve this inconsistency on the fate potentials of injury-reactivated tectal RG.

In this study, we set out to investigate the mechanism controlling injury responses of tectal RG in vivo. We found that stab injury drove a subset of tectal RG into the cell cycle. Surprisingly, proliferative tectal RG responding to the sequential injuries at the same injury site were distinct but overlapping. Quantitative analysis showed the probability of proliferative RG responding to both sequential stab injuries could be well explained by a model incorporating stochastic cell-cycle entry at the fixed probability of ~25%. Single-cell RNA-seq and functional analysis revealed this stochastic cell-cycle entry was dependent on Notch/Delta lateral inhibition. The clonal analysis showed that proliferative tectal RG underwent gliogenesis. Interestingly, post-injury notch inhibition drove reactive RG into

the cell cycle and resulted in increased neurogenesis. Interestingly, the over-produced neurons mostly diminished by approximately 25 days post-injury (dpi).

## Results

### Stab injury induces the proliferation of dormant tectal RG

Consistent with earlier studies (*Galant et al., 2016*; *Ito et al., 2010*; *Jung et al., 2012*), our results showed that Tg(*gfap*:GFP) (*Bernardos and Raymond, 2006*), Tg(*her4.1*:dRFP) (*Yeo et al., 2007*), and the antibody against glutamine synthetase (GS) specifically labeled RG that line the tectal ventricle and extend their basal processes into the superficial neuropils (*Figure 1A–B₃* and *Figure 1—figure supplement 1A-C₂*). To examine the dormancy of tectal RG, we quantified the proliferative RG in the optic tectum of Tg(*gfap*:GFP) fish using the antibodies against proliferating cell nuclear antigen (PCNA), glutamine synthetase (GS), and GFP (*Figure 1A–B₃*). Only $1.4 \pm 0.2\%$ (n = 5, mean $\pm$ SEM) of GFP$^+$/GS$^+$ RG at the bottom of the periventricular gray zone (PGZ) were PCNA$^+$ (*Figure 1A–B₃*). Bromodeoxyuridine (BrdU) is a nucleotide analog that incorporates into new synthesized DNA of dividing cells in the S phase. We noticed that 1 day's administration of BrdU labeled very few tectal RG under physiological conditions (*Figure 1—figure supplement 1D-F₁*). Together, these results demonstrate that zebrafish tectal RG are largely dormant under normal physiological conditions.

Next, we set out to investigate whether injury induces RG proliferation. We applied a stab injury to the central-dorsal part of the optic tectum (*Figure 1C*). At 3 dpi, immunostaining showed marked expression of PCNA in some RG underneath the injury site but not in the uninjured control hemisphere (*Figure 1D–G*). BrdU incorporation and staining experiments showed that injury-reactivated RG were labeled by BrdU at 3 dpi, indicating that dormant RG entered S phase after injury (*Figure 1—figure supplement 2A–C*).

To confirm the proliferation of injury-reactivated RG, we took advantage of the Cre-loxP system to do the long-term clonal analysis of single tectal RG (*Kroehne et al., 2011*). We used the Tg(*her4.1*:mCherryT2ACreER$^{T2}$) transgenic line, in which the promoter of *her4.1* drives the expression of the mCherry fluorescent protein and CreER$^{T2}$ recombinase in tectal RG (*Figure 1H*). By crossing this line with Tg(*hsp70l*:DsRed2(*floxed*)EGFP) (*Figure 1H*), individual RG in the uninjured optic tectum can be genetically labeled by tamoxifen (TAM) administration (at 2 to 3 months old) and the resulting clones visualized by EGFP after heat shock at the desired time points (*Figure 1—figure supplement 2D and E*). We combined the Cre-loxP system and (5-ethynyl-2′-deoxyuridine) EdU pulses for six consecutive days after the injury to perform the clonal analysis of single RG after stab injury (*Figure 1I*). Final clones were analyzed at desired time points after a pulse of EGFP expression by heat shock (*Figure 1I*). At 8 dpi, we observed marked RG derived clones (EGFP$^+$/EdU$^+$, *Figure 1J–K₃*). In total 29 clones were collected (from 11 fish), of which ~ 69% were 2 cell clones (20/29; *Figures 1M–M2, O and O1*),~24% were 1 cell clones (7/29; *Figures 1L–L2, O and O1*), and the rest were 3 cell clones (2/29; *Figure 1N–O₁*). Taken together, dormant tectal RG are capable of proliferation after stab injury.

We then examined the injury responses of RG in different geographical regions of the optic tectum. Stab injury was applied at five different regions, including the anterior-dorsal, central-dorsal (as positive control region), medial-dorsal, lateral, and posterior-dorsal regions of the right hemisphere (*Figure 1—figure supplement 2F–O*). We stained and quantified the PCNA$^+$ RG underneath the injury sites in all regions, showing that the number of PCNA$^+$ RG was not significantly different across regions except for the medial-dorsal region, where the RG had little proliferative capacity at 3 dpi (*Figure 1—figure supplement 2H,M and P*). To confirm this, we performed two-sites injury on the same hemisphere of the optic tectum, the medial-dorsal region, and the central-dorsal region were injured (*Figure 1—figure supplement 2Q–S*). Consistently, significantly more PNCA$^+$ RG were in the central-dorsal region than in the medial-dorsal region (*Figure 1—figure supplement 2T*). Together, our results indicate that stab injury can induce RG proliferation across distinct regions in the optic tectum except in the medial-dorsal region.

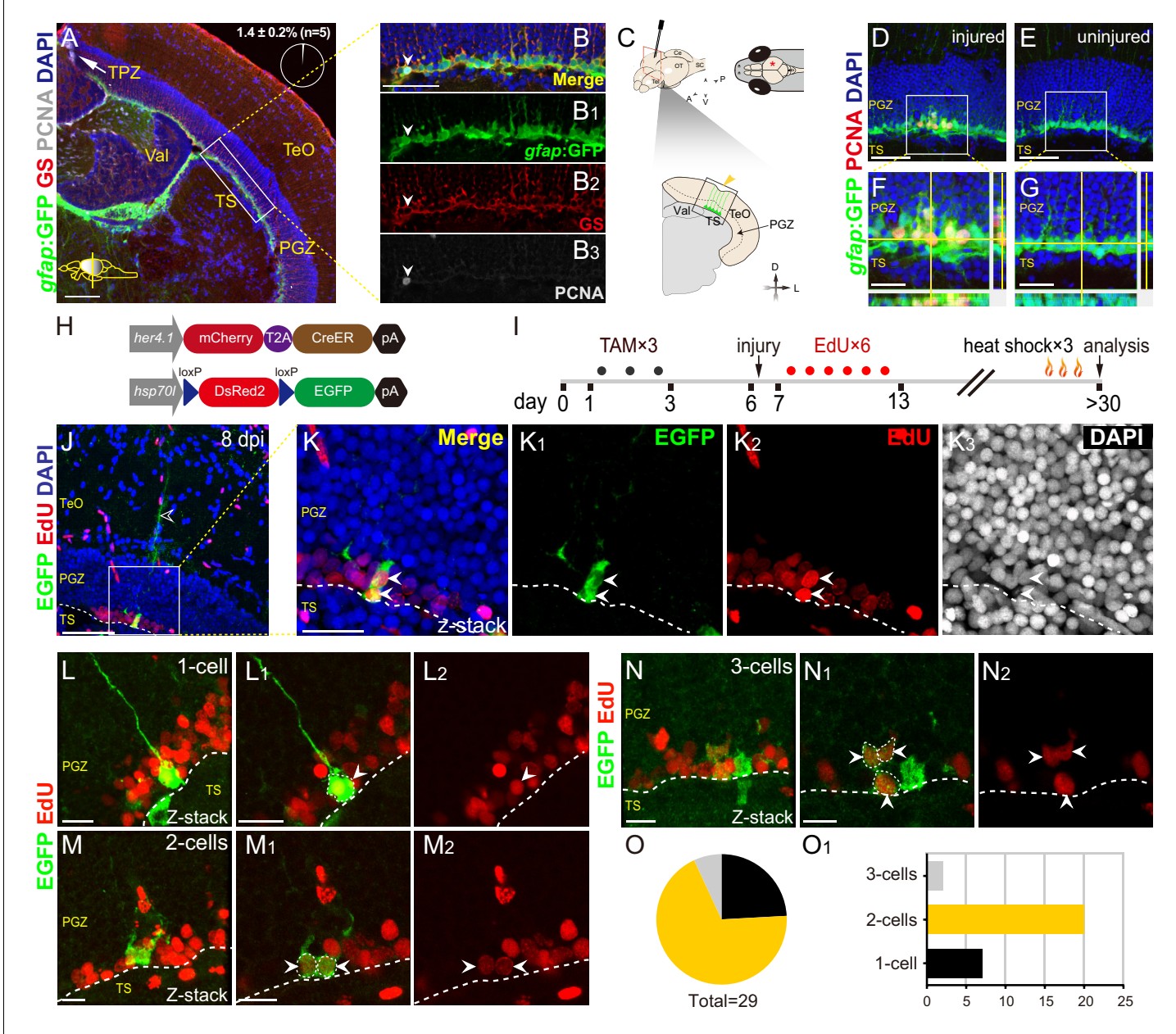

**Figure 1.** Injury reactivates dormant RG to proliferate and divide. (**A–B₃**) Tg(*gfap*:GFP) (green), GS (red) and PCNA (gray) immunofluorescences show that PCNA⁺ proliferative cells (gray cells) are restricted to the TPZ (white arrow in (**A**)) and very few radial glia (RG) (1.4 ± 0.2%, n = 5, mean ± SEM, gray cells, white arrowheads in (**B–B₃**)) is PCNA⁺. (**B–B₃**) The high-magnification images of the boxed area (white box) in (**A**). (**C**) Schematic representation of stab injury assay. A 30G needle is stabbed into the central-dorsal region of the right hemisphere of zebrafish optic tectum. The red asterisk and yellow arrowhead indicates the injury site. RG (green cells) at the bottom of PGZ underneath the injury site are analyzed. (**D–G**) Tg(*gfap*:GFP) (green) and PCNA (red) immunofluorescences show that injury induces the proliferation of RG (GFP⁺/PCNA⁺, yellow cells) underneath the injury site at 3 days post-injury (dpi). (**F and G**) The high-magnification images of boxed areas in (**D**) and (**E**), respectively. (**H**) The design of Cre-loxP transgenic fish lines used for clonal analysis of individual tectal RG. Fish expressing mCherryT2ACreER^T2 controlled by the *her4.1*-promotor are crossed to red-to-green reporter fish controlled by the *hsp70l* promoter. In Tg(*her4.1*:mCherry-CreER^T2 × *hsp70l*:DsRed2(*floxed*)EGFP) double transgenic fish, EGFP expression is specifically induced in *her4.1*-expressing RG and their progeny by TAM applications and heat shocks. (**I**) Experimental time course of Cre-loxP-based clonal analysis experiments shown in (**J–O₁**). Double transgenic fish are administrated with TAM for three consecutive days (black dots) before the injury. EdU is injected to the injured fish to label the newborn cells for six consecutive days (red dots). Fish (21 to 24 dpi) are heat-shocked to induce EGFP expression in recombined cells and their progeny. (**J–K₃**) Representative RG-derived clone (EGFP⁺/EdU⁺, white arrows) underneath the injury site at 8 dpi. (**K–K₃**) The high-magnification images of the boxed area in (**J**). Two EGFP⁺/EdU⁺ (white arrowheads) cells and an EGFP⁺ radial process (open white arrowhead in (**J**)) are found underneath the injury site in this clone. (**L–N₂**) Representative 1 cell (**L–L₂**), 2 cells (**M–M₂**) and 3 cells clones (**N–N₂**) derived from single

*Figure 1 continued on next page*

*Figure 1 continued*

RG in response to the stab injury. In these clones, cells are EGFP$^+$/EdU$^+$ newborn cells (white arrowheads). (**O and O$_1$**) The size distribution of collected 29 clones. 2-cells clones (20/29) are the most abundant clones. White dashed lines represent the tectal ventricle boundary. A, anterior; P, posterior; D, dorsal; V, ventral; L, lateral; Tel, telencephalon; OT, optic tectum; Ce, cerebellum; SC, spinal cord; TAM, tamoxifen; RG, radial glia; TeO, tectum opticum; TPZ, tectal proliferation zone; PGZ, periventricular gray zone; TS, torus semicircularis; Val, valvula cerebelli. Scale bars, 100 μm (**A**); 50 μm (**B**-B$_3$, D-G, and J); 20 μm (**K**–K$_3$); and 10 μm (**L**–N$_2$).

DOI: https://doi.org/10.7554/eLife.48660.003

The following figure supplements are available for figure 1:

**Figure supplement 1.** Tectal RG are largely dormant under physiological conditions.

DOI: https://doi.org/10.7554/eLife.48660.004

**Figure supplement 2.** Injury responses of RG in different geographical regions in the optic tectum.

DOI: https://doi.org/10.7554/eLife.48660.005

## Injury-reactivated tectal RG enter the cell cycle stochastically

To investigate the cell-cycle entry of tectal RG, we examined PCNA expression in tectal RG of Tg (*1016tuba1α*:GFP), a transgenic line used as the reporter for retinal MG reactivation after the injury (*Fausett and Goldman, 2006*). Under physiological conditions, weak GFP signals were present in tectal RG (*Figure 2A and F*). At 1 dpi, robust GFP signals were already observed together with the upregulation of Vimentin, a hallmark of glial reactivation at the early stage of gliosis (*Figure 2—figure supplement 1A*-B$_1$) (*Liddelow and Barres, 2017*), whereas only few of RG was PCNA$^+$ (*Figure 2B,G* and *Figure 2—figure supplement 1C*), suggesting a lack of proliferation. The number of reactive RG, the ones with robust GFP signals, peaked at 3 dpi (*Figure 2C,H and K*), while a significant number of PCNA$^+$ RG first occurred at 2 dpi and peaked at approximately 3–4 dpi (*Figure 2K* and *Figure 2—figure supplement 1C*). From 4 to 7 dpi, the number of PCNA$^+$ RG gradually dropped back to the same level observed before the injury (*Figure 2K* and *Figure 2—figure supplement 1C*). At 3 and 5 dpi, we found some robust GFP signals and PCNA$^+$ cells at the injury site and in the region underneath (*Figure 2C and D*).The GFP signals were likely due to hypertrophic responses of RG's processes and other cells at the injury site (*Figure 2D*), while PCNA$^+$ cells likely consisted of oligodendrocytes precursor cells, recruited microglia/macrophage and other cell types (*Figure 2—figure supplement 1D-H$_3$*).

We further measured the spatial relation between reactive RG (with robust GFP$^+$ signal) and proliferative RG (GFP$^+$/PCNA$^+$) (*Figure 2L*). In coronal sections, reactive RG were primarily distributed in an area with a width of 186 ± 4 μm (n = 7, mean ± SEM) underneath of the injury sites ('Reactive Zone') while the majority of proliferative RG (88 ± 3%, n = 7, mean ± SEM) were located in an area with the width of 76 ± 5 μm (n = 7, mean ± SEM) underneath of the injury site ('Proliferative Zone') (*Figure 2L and M*). The small variation of the width of both zones indicated the high reproducibility of stab injury outcomes (*Figure 2M*).

Although the injury reactivated all RG underneath the injury site, only a subset of them (~25%, n = 8) became proliferative (*Figure 2H* and *Figure 2—figure supplement 1C*). It raised an immediate question as to whether the proliferation of a subset of RG was due to stochastic cell-cycle entry or the presence of distinct RG subpopulations that respond differentially to the stab injury. To test this, we designed a sequential stab injury experiment. We examined the responses of reactive tectal RG to two sequential stab injuries performed at the same physical site (*Figure 2N*). The first injury was introduced followed by BrdU pluses for six consecutive days to label proliferative RG responding to the first injury, and the second injury was introduced at 12 dpi followed by EdU pluses for six consecutive days to mark proliferative RG responding to the second injury (*Figure 2N*). Finally, the fish were sacrificed, and coronal sections were stained for BrdU, EdU, radial glial marker BLBP, and neuronal marker HuC/D at 23 dpi (*Figure 2O–O$_5$*). We found that although the number of proliferative RG induced by the first and the second injury showed no significant difference (the first injury: 84.4 ± 15.0 cells, n = 8; the second injury: 83 ± 9.4 cells, n = 8; mean ± SEM; p>0.05; *Figure 2P*), two sets of proliferative RG were distinct with some degree of overlapping (23 ± 6, n = 8, mean ± SEM; *Figure 2O–O$_5$* and *Figure 2P*). More importantly, the proportion of overlapping RG (those reactivated after both injuries) was statistically indistinguishable from the multiplication of the reactivation probabilities of either injury, which suggested that individual reactive RG entered the

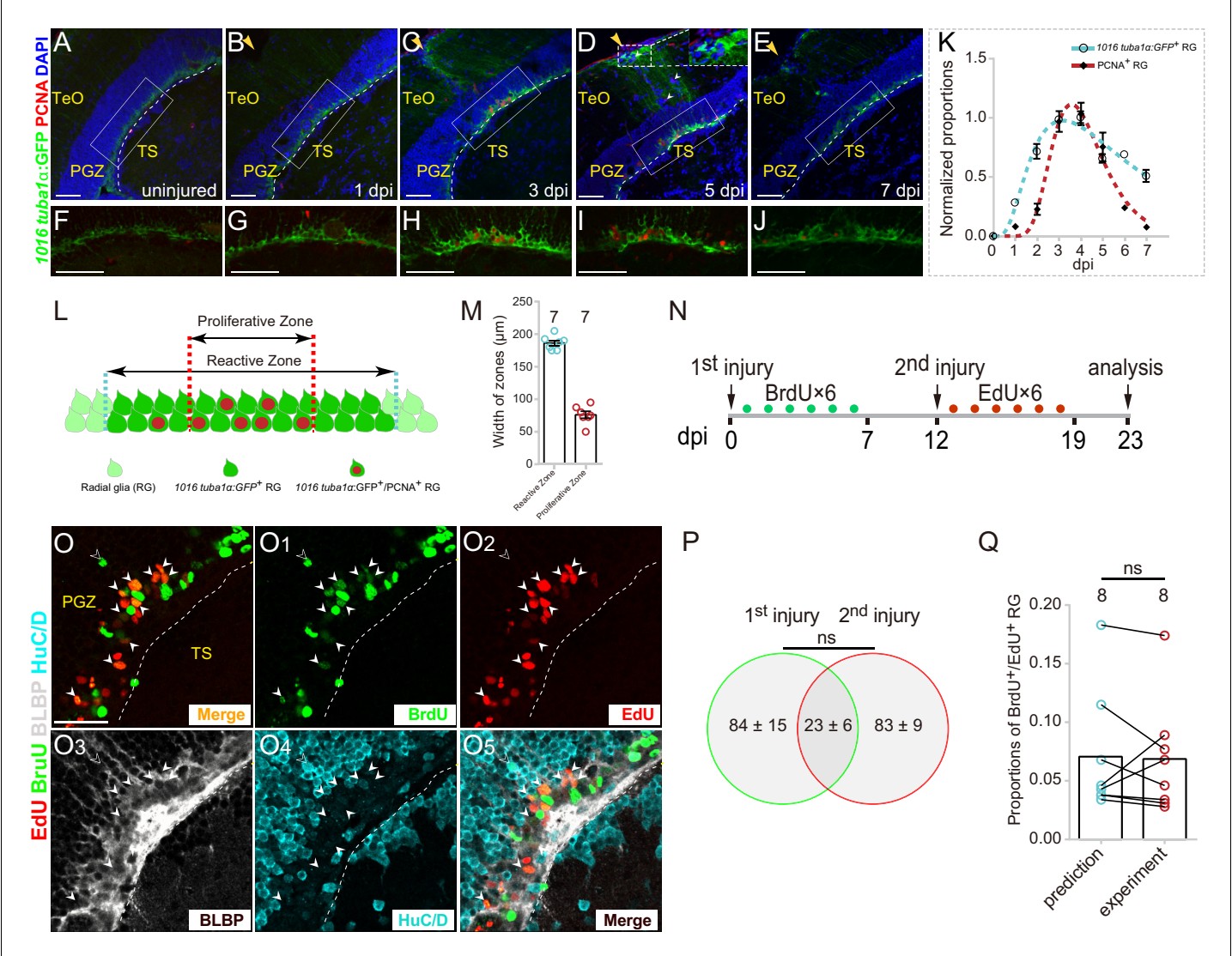

**Figure 2.** Injury-reactivated RG enter the cell cycle in a stochastic manner. (A–J) Low- and high-magnification immunofluorescence images showing the dynamics of GFP and PCNA of RG underneath the injury sites (yellow arrows) in the optic tecta of the Tg(*1016tuba1α*:GFP) fish throughout 0 to 7 dpi. At 3 and 5 dpi, GFP is strongly upregulated in the cell body and radial processes (white dashed box in (D)). Some GFP⁻/PCNA⁺ cells (white arrowheads in (D)) are non-RG proliferative cells induced by injury. (F–J) The high-magnification images of boxed areas in (B–E). See also *Figure 2—figure supplement 1D-H₃*. (K) Dot plots showing the change of normalized proportions of GFP⁺ (open circles, the cyan dashed line representing a fitting curve) and PCNA⁺ (solid diamonds, the red dashed line representing a fitting curve) RG throughout 0 to 7 dpi. The numbers are fitted by lognormal nonlinear regression. (n = 3, 3, 3, 3, 3, 3, 3, 3 for GFP; n = 3, 4, 4, 10, 9, 3, 3, 2 for PCNA). See also *Figure 2—figure supplement 1C*. (L and M) Schematic representation and quantification of the physical distribution of Reactive and Proliferative Zones in the injured optic tectum of Tg (*1016tuba1α*:GFP) fish. GFP is upregulated in RG in Reactive Zone and Proliferative Zone, 88 ± 3% (n = 7, mean ± SEM) proliferative RG (PCNA⁺/GFP⁺) are located in Proliferative Zone. (N) Experimental time course of the sequential injury experiment shown in (O–O₅). Six consecutive days of BrdU (green dots) are injected after the first injury, and the second injury was applied at 12 dpi followed by EdU injections for six consecutive days (red dots), and finally, fish were analyzed at 23 dpi. (O–O₅) BLBP (gray), HuC/D (cyan), BrdU (green) and EdU (red) immunofluorescences show that proliferative RG that respond to the stab injury of either the first time (BrdU⁺/BLBP⁺, green cells) or the second time (EdU⁺/BLBP⁺, red cells) are distinct but overlapping. RG that respond to both injuries are BrdU⁺/EdU⁺/BLBP⁺ (yellow cells) indicated by white arrowheads. The open white arrowheads indicate a newborn neuron (BrdU⁺/HuC/D⁺) generated from the RG respond to the first injury. Venn diagram of the number of RG that enter the cell cycle in a sequential injury experiment. No significant difference is shown between the number of RG induced by the first and the second injury (mean ± SEM; ns, p>0.05; Wilcoxon test). The predicted and experimental proportions of RG entering cell cycle in response to the injury of both times showing no significant difference. The prediction is derived from the multiplication of the reactivation probabilities of either injury, with the assumption of stochastic cell-cycle entry of reactive RG (mean ± SEM; ns, p>0.05; Wilcoxon test). The numbers above the bars indicate the animals used. White dashed lines represent the tectal ventricle boundary. RG, radial glia; TeO, tectum opticum; PGZ, periventricular gray zone; TS, torus semicircularis. Scale bars, 50 µm (A–J); and 30 µm (O–O₅). See also *Figure 2—figure supplement 1*.

*Figure 2 continued on next page*

*Figure 2 continued*

DOI: https://doi.org/10.7554/eLife.48660.006

The following source data and figure supplement are available for figure 2:

**Source data 1.** Quantification of the number of RG that enter the cell cycle in the sequential injury experiment.

DOI: https://doi.org/10.7554/eLife.48660.008

**Source data 2.** The predicted and experimental proportions of RG entering cell cycle in response to the injury of both times.

DOI: https://doi.org/10.7554/eLife.48660.009

**Figure supplement 1.** Injury induces proliferation of other cell types.

DOI: https://doi.org/10.7554/eLife.48660.007

cell cycle in the stochastic manner (prediction: $7.1 \pm 1.9\%$, n = 8; experiment: $6.8 \pm 1.7\%$, n = 8; mean ± SEM; p>0.05; *Figure 2Q*).

## Single-cell RNA-seq analysis reveals cellular states representing RG reactivation and proliferation

To further examine the molecular mechanism underlying this stochastic cell-cycle entry of injury-reactivated RG, we carried out single-cell RNA sequencing (scRNA-seq) analysis of tectal RG at 3 dpi, at which stage the number of proliferative tectal RG nearly reached the plateau in terms of cell number (*Figure 2K* and *Figure 2—figure supplement 1C*). We dissected and dissociated the optic tecta of Tg(*gfap*:GFP) fish at 3 dpi and sorted out GFP$^+$ RG using fluorescence-activated cell sorting (FACS) for further scRNA-seq on the 10x Genomics platform (*Figure 3A* and *Figure 3—figure supplement 1A-1A$_2$*). The gene profiles of in total 2998 single cells were qualified after the initial filtering using the Seurat algorithm (*Figure 3—figure supplement 1B and C*; see also details in Materials and Methods; http://satijalab.org/seurat/). We performed raw cell clustering using t-stochastic neighbor embedding (t-SNE) analysis (*Figure 3—figure supplement 1B and C*). According to cell-type-specific genes, we excluded the cell clusters representing non-glial cells (*Figure 3—figure supplement 1D,E*), RG in the tectal proliferation zone (TPZ), and oligodendrocytes (*Figure 3—figure supplement 2A–F*; see also details in Materials and Methods). The remaining 1174 cells exhibited radial glial characteristics and were thus used for further analysis. They were segregated into five major cell clusters using t-SNE analysis (*Figure 3B*). Each cell cluster had a characteristic gene expression (*Figure 3C*). Cluster 1 cells (RG of dormant state, dRG) constituted the most abundant cell population with the high expression of milk-fat globule-epidermal growth factor 8a (*mfge8a*, *Figure 3D*), whose analog, *mfge8*, is a phagocytosis factor that maintains the pool of radial glia-like cells by controlling cellular quiescence in mice (*Zhou et al., 2018*). Cluster 2 cells (RG of reactive state) were characterized by their up-regulation of vimentin (*vim*), a hallmark of RG reactivation (*Figure 3E* and *Figure 2—figure supplement 1A-B$_1$*). Proliferative RG were composed of RG of proliferative-S (*mcm2* and *pcna*, cluster 3; *Figure 3F and G*) and proliferative-G2 states (*cdk1* and *nusap1*, cluster 4; *Figure 3H and I*). Cluster 5 cells highly expressed vimentin (*vim*) and were likely to represent Vimentin$^+$ cells from neighboring tissues under the optic tectum in the midbrain due to possible contamination during the dissection of the optic tecta (*Figure 3E* and *Figure 3—figure supplement 2G-2I$_1$*). We did not observe such a high expression of *vim* in the tectal RG (*Figure 3E*). Thus, we excluded cluster 5 cells from further analysis. Cell cycle phases analysis (*Figure 3J*) and pseudo-time analysis (*Figure 3K* and *Figure 3—figure supplement 2J*) were performed and suggested the temporal order of 4 remaining cell clusters, thereafter termed as the state of dormant RG (dRG), the state of reactive RG (reactive RG), the state of proliferative-S RG and the state of proliferative-G2 RG.

Next, we looked into the expression dynamics of the genes that differentially expressed across the states. *mfge8a* was abundant in dormant RG (cluster 1), began to decrease in reactive RG (cluster 2) and became rapidly diminished in proliferative RG (cluster 3 and 4) (*Figure 3L*). Kruppel-like transcription factor 6a (*klf6a*), the transcription factor essential for optic axon regeneration (*Veldman et al., 2010*; *Veldman et al., 2007*), exhibited a peaked expression in reactive RG (cluster 2) (*Figure 3M*), and Insulinoma-associated 1a (*insm1a*), encoding a transcriptional repressor that has been reported to be necessary for MG-based retina regeneration (*Forbes-Osborne et al., 2013*; *Ramachandran et al., 2012*), highly expressed in proliferative-S and -G2 RG (cluster 3 and 4)

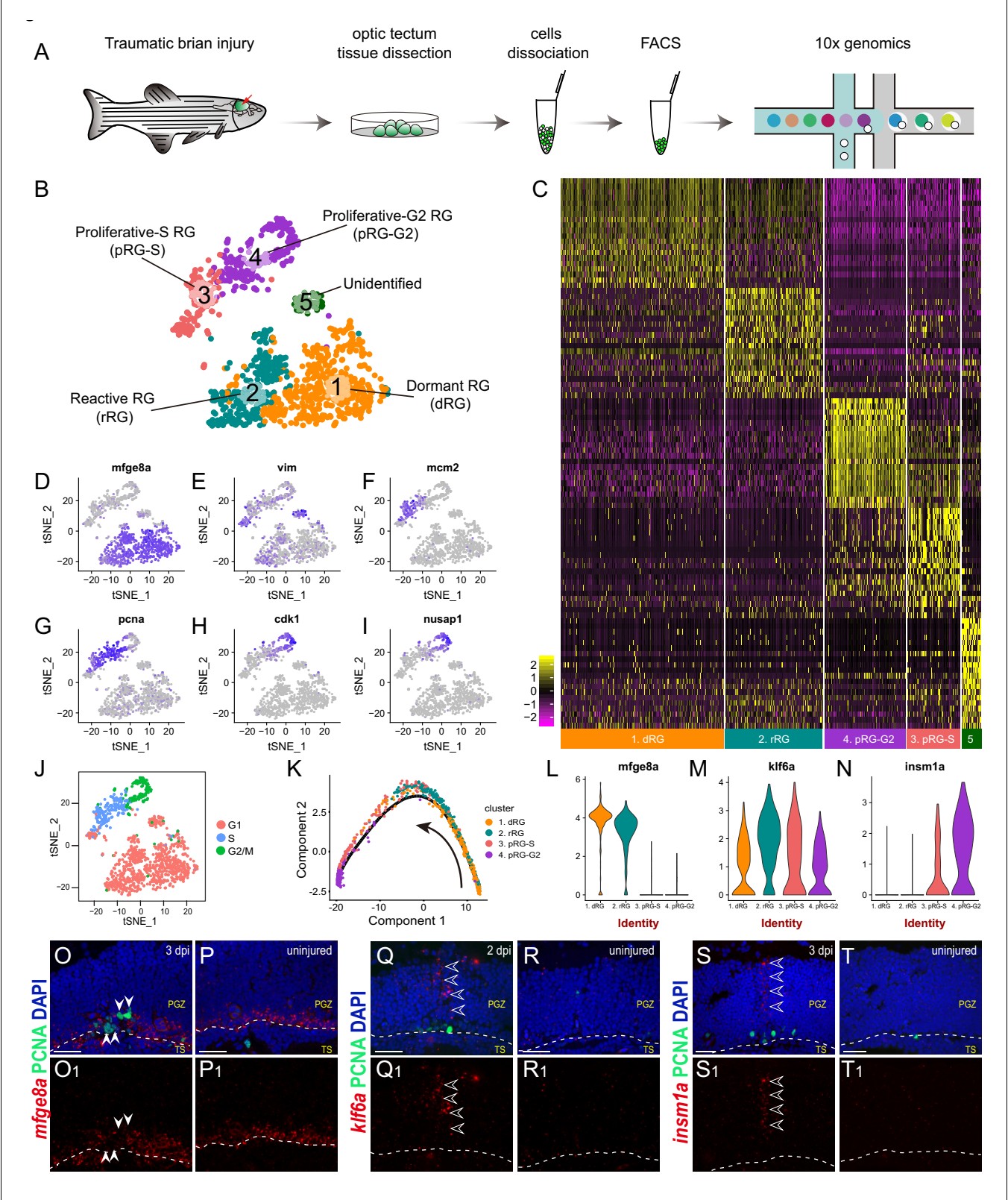

**Figure 3.** Single-cell RNAseq revealing cellular states underlying the cell-cycle entry of reactive RG. (A) Workflow for single-cell RNA-seq (scRNA-seq) of tectal RG after stab injury. Optic tecta are dissected from 3 dpi Tg(*gfap*:GFP) zebrafish brain and dissociated into a single-cell suspension. Single GFP⁺ RG are sorted by fluorescence-activated cell sorting (FACS) and followed by 10x genomics scRNA-seq. (B) A t-SNE plot of 1174 single tectal RG at 3 dpi revealing 5 cell clusters. Dormant RG (dRG, cluster 1) in orange; Reactive RG (rRG, cluster 2) RG in dark cyan; Proliferative-S RG (pRG-S, cluster 3) in

*Figure 3 continued on next page*

*Figure 3 continued*

Indian red; Proliferative-G2 RG (pRG-G2, cluster 4) in purple; Unidentified RG (cluster 5) in dark green. (**C**) Heatmap showing the expression of the top 20 marker genes that characterize each cell clusters. Rows represent genes while columns represent cells. (**D–I**) t-SNE plots showing expression of state-specific genes of distinct cell states. (**J**) Cell-cycle characteristics of individual cell states. S phase-related genes are mainly expressed in pRG-S cluster (cluster 3), G2/M-related genes are mainly expressed in pRG-G2 cluster (cluster 4). (**K**) Pseudo-time developmental trajectory of identified states using Monocle showing that the trajectory is booted from dRG cluster (cluster 1) and end at pRG-G2 cluster (cluster 4). (**L–N**) Violin plots of expression for genes enriched in dRG cluster (*mfge8a*, cluster 1), rRG cluster (*klf6a*, cluster 2) and pRG-S/G2 cluster (*insm1a*, cluster 3 and 4). (**O–T₁**) In situ hybridization showing the expression of *mfge8a* (**O–P₁**), *klf6a* (**Q–R₁**) and *insm1a* (**Q–R₁**) in the optic tecta after injury. The white arrowheads shown in (**O and O₁**) indicate PCNA⁺ proliferative RG are *mfge8a⁻*, the open white arrowheads indicate *klf6a* (**Q and Q₁**) or *insm1a* (**S and T₁**) mRNA signals are located in processes of proliferative RG. White dashed lines represent the tectal ventricle boundary. t-SNE, t-stochastic neighbor embedding; RG, radial glia; PGZ, periventricular gray zone, TS, torus semicircularis. Scale bars, 30 μm. See also *Figure 3—figure supplements 1* and *2* and Materials and methods.

DOI: https://doi.org/10.7554/eLife.48660.010

The following figure supplements are available for figure 3:

**Figure supplement 1.** Glial and Non-glial cell clusters identification from the scRNA-seq data.

DOI: https://doi.org/10.7554/eLife.48660.011

**Figure supplement 2.** Identification of the clusters representing RG in the TPZ and oligodendrocytes.

DOI: https://doi.org/10.7554/eLife.48660.012

(*Figure 3N*). To verify their expression, we performed in situ hybridization. The results were consistent with our scRNA-seq data, *mfge8a* was down-regulated in injured-induced PCNA⁺ proliferative RG at 3 dpi (*Figure 3O–P₁*), whereas *klf6a* and *insm1a* mRNA expression increased in the 2-dpi (*Figure 3Q–R₁*) and 3-dpi (*Figure 3S–T₁*) optic tecta, respectively. Interestingly, the signals of *klf6a* (*Figure 3Q and Q₁*) and *insm1a* (*Figure 3S and S₁*) were mainly distributed in the processes of RG.

## Notch/Delta expression pattern correlated with the cell-cycle entry of reactive RG

Notably, during the transition of reactive (cluster 2) and proliferative states (cluster 3 and 4), the expression of *her4.1*, the targeting gene of Notch signaling (*Takke et al., 1999*), decreased (*Figure 4A and B*), whereas *deltaA* expression increased (*Figure 4C and D*). Further correlation analysis showed that *pcna* and *deltaA* expression were correlated, while *pcna and deltaA* were uncorrelated with the expression of *her4.1* and *her4.2* (*Figure 4E*). Our results suggest proliferative RG with an increase of *deltaA* expression and a decrease of Notch activity.

To visualize the Notch/Delta dynamics in vivo, we employed a reporter line Tg(*Tp1bglob*:EGFP) (hereafter referred to as Tg(*Tp1*:EGFP)), in which EGFP is driven by the TP1 element, the direct target of the intracellular domain of Notch receptors (NICD) that is generated upon Notch activation (*Parsons et al., 2009*; *Quillien et al., 2014*). We performed PCNA immunostaining on the coronal sections of Tg(*Tp1*:EGFP) at 3 dpi (*Figure 5F–G₃*). Interestingly, the results showed ~82% (97/119 cells, n = 6 sections) of PCNA⁺ proliferative RG had no EGFP signal, indicating low Notch activity (*Figure 4H*). Notch activity and PCNA signal were mostly exclusive (*Figure 4F₁–F₃*). Consistently, in situ hybridization of *deltaA* followed by immunostaining of PCNA showed ~81% (60/74 cells, n = 10 sections) of PCNA⁺ RG expressed *deltaA* (*Figure 4I–4J*). Our results suggest that Notch/Delta lateral inhibition may be at work.

## Notch inhibition mediates stochastic cell-cycle entry of reactive RG

As Notch/Delta lateral inhibition contributes to the mosaic entry of embryonic neurogenesis of neural progenitor cells (*Cabrera, 1990*; *Dong et al., 2012*; *Formosa-Jordan et al., 2013*; *Kageyama et al., 2008*; *Sato et al., 2016*; *Tiedemann et al., 2017*), we examined its role in the cell-cycle entry of injury-reactivated RG. We disturbed Notch/Delta lateral inhibition by blocking Notch signaling using LY411575, a potent inhibitor of the γ-secretase complex, which acts by preventing the cleavage of NICD (*Figure 5A*) (*Geling et al., 2002*; *Katz et al., 2016*). Notably, 2 days' LY411575 treatment resulted in a ~ 4 folds increase in the number of proliferative RG in the injured optic tectum (DMSO-treated: 24.8 ± 3.7 cells, n = 6; LY411575-treated: 109 ± 16.9 cells, n = 5; mean ± SEM; ***p<0.001; *Figure 5B, C, F, G, and J*). RO4929097, another Notch signaling inhibitor, resulted in a similar phenotype (*Figure 5—figure supplement 1A–D*). Consistent with the findings

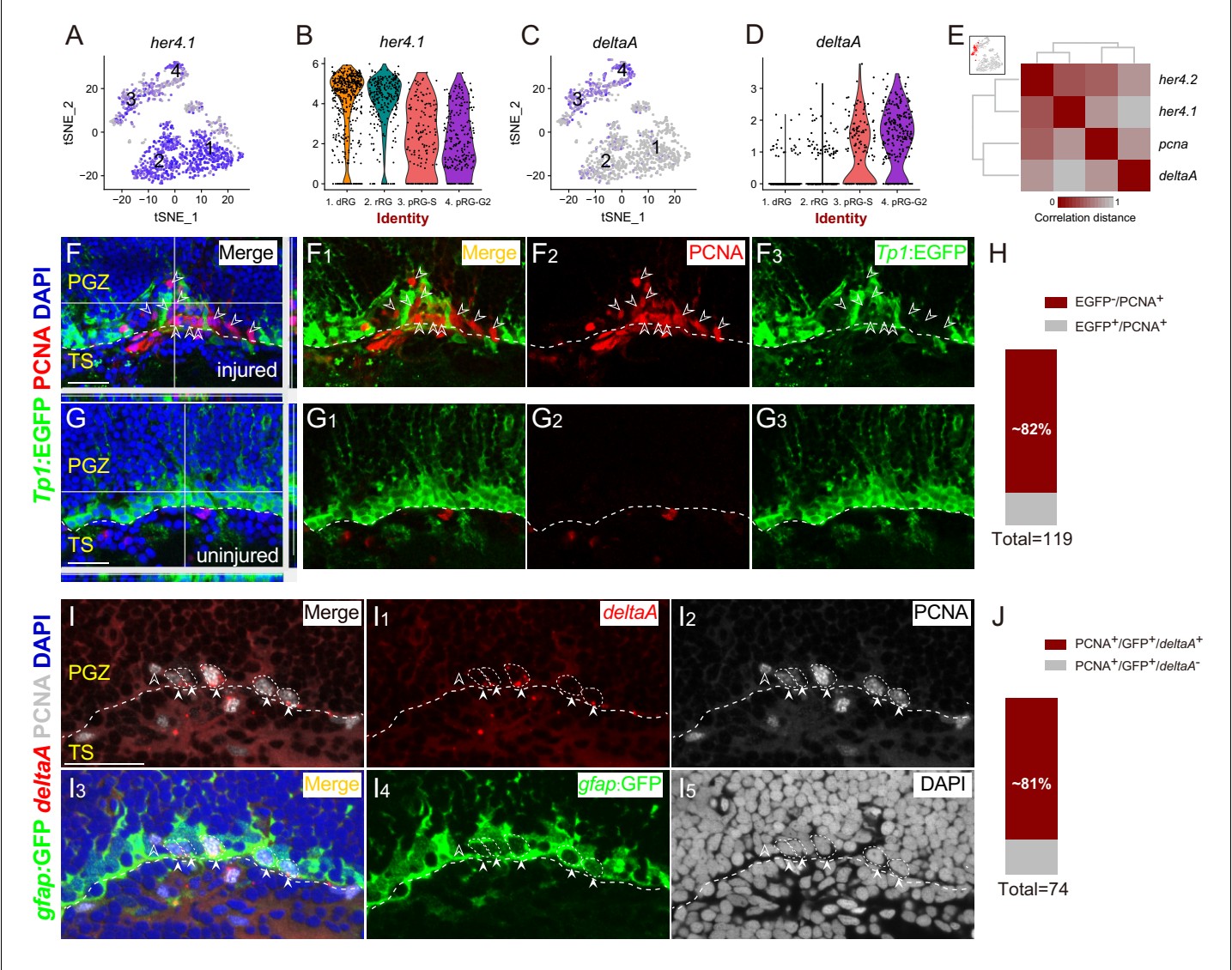

**Figure 4.** Spatial Distribution of Notch and Delta in reactive RG after the injury. (**A–D**) t-SNE plots (**A and C**) and violin plots (**B and D**) showing *deltaA* is mainly expressed in pRG-S (cluster 3) and pRG-G2 (cluster 4) RG, whereas the expression of *her4.1*, a target gene of Notch signaling, is down-regulated in pRG-S (cluster 3) and pRG-G2 RG (cluster 4). (**E**) Correlation distance matrix showing *deltaA* exhibits the high correlation with *pcna* but the low correlation with *her4.1* and *her4.2* in pRG-S RG (cluster 3, red dots in the t-SNE plot at the top left, each dot represents a single cell). (**F–G$_3$**) Tg (*Tp1*:EGFP) (green) and PCNA (red) immunofluorescences show that PCNA$^+$ proliferative RG (open white arrowheads, red cells) and EGFP$^+$ RG (green cells) are exclusive in the injured optic tectum (**F–F$_3$**). (**G–G$_3$**) The representative images of the uninjured optic tectum. (**H**) Quantification of EGFP$^-$/PCNA$^+$ and EGFP$^+$/PCNA$^+$ RG in (**F**) showing ~82% PCNA$^+$ RG are EGFP$^-$. Most of the PCNA$^+$ proliferative RG have low Notch activity (97/119 cells in 6 sections from 4 fish). (**I–I$_5$**) Specific expression of *deltaA* in PCNA$^+$ tectal RG (white arrowheads, white dashed circles) at 3 dpi is confirmed by in situ hybridization. The open white arrowheads indicate a GFP$^-$/PCNA$^+$/*deltaA$^-$* non-RG proliferative cells. (**J**) Quantification of PCNA$^+$/*deltaA$^+$* and PCNA$^+$/*deltaA$^-$* RG in (**I**) showing ~81% PCNA$^+$ RG express *deltaA*. (60/74 cells in 10 sections from 5 fish). White dashed lines represent the tectal ventricle boundary. RG, radial glia; PGZ, periventricular gray zone; TS, torus semicircularis. Scale bars, 30 μm.

DOI: https://doi.org/10.7554/eLife.48660.013

of a recent study (*Ueda et al., 2018*), Notch inhibition was also sufficient to trigger the proliferation of tectal RG even without any injury (DMSO-treated: 6.3 ± 0.48 cells, n = 4; LY411575-treated: 26.7 ± 2.4 cells, n = 5; mean ± SEM; p>0.05; *Figure 5D, E, H, I, J*, *Figure 5—figure supplement 1C and D*), which was reminiscent of the increase of constitutively proliferative RG in the zebrafish telencephalon by Notch inhibition (*Chapouton et al., 2010*).

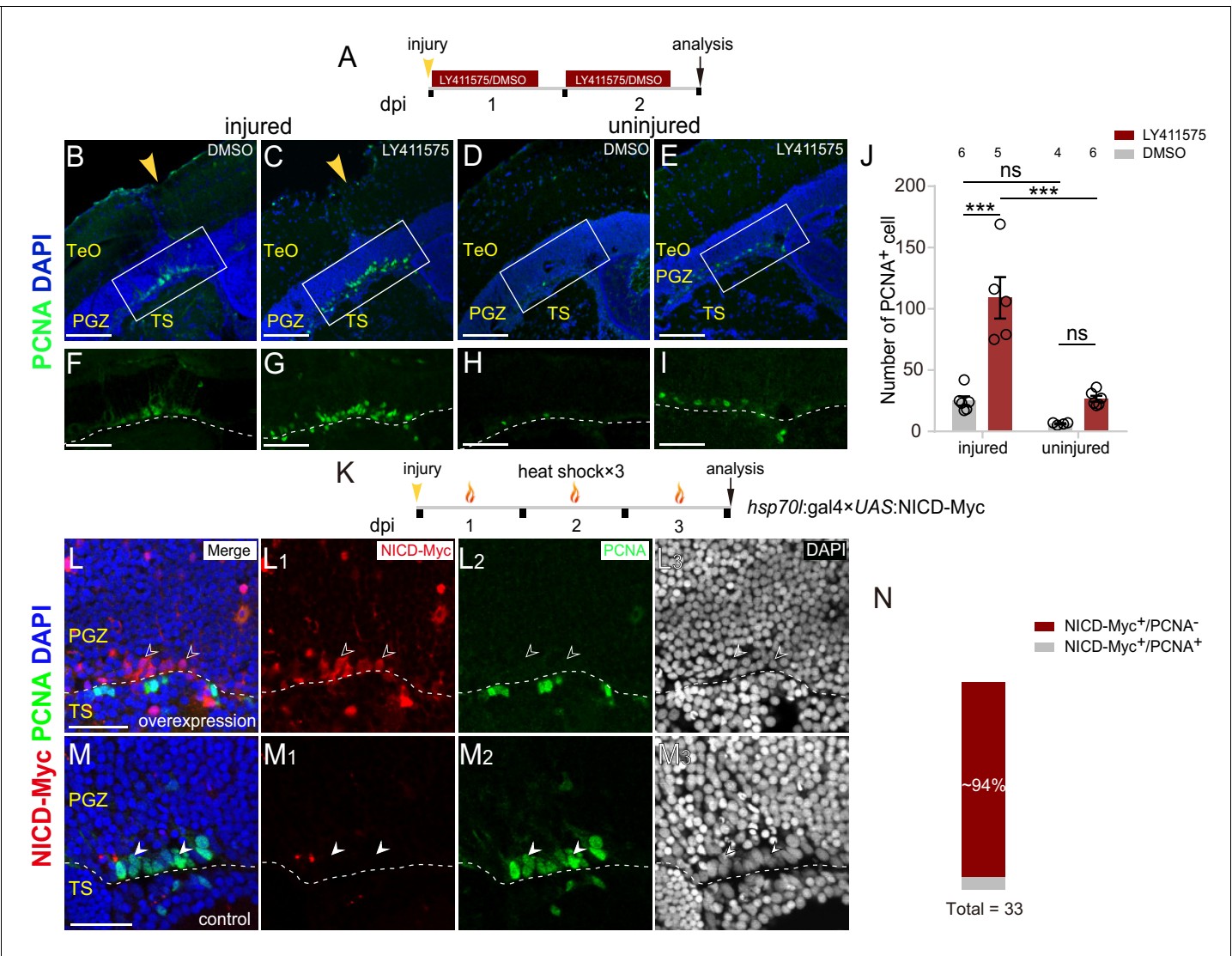

**Figure 5.** Notch inhibition mediates the proliferation of reactive tectal RG. (**A**) Experimental time course of Notch inhibition experiments shown in (**B–I**). Fish are administrated with LY411575, a Notch inhibitor, or DMSO for two consecutive and are analyzed at 2 dpi. (**B–I**) LY411575 administration increases the number of proliferative RG (PCNA$^+$, green cells) underneath the injury site in the optic tectum with (**B and C**) or without injury (**D and E**). (**F–I**) The high-magnification images of boxed areas in (**B–E**). (**J**) Quantification of PCNA$^+$ cell in (**B–E**). LY411575 administration significantly increases the number of proliferative RG (PCNA$^+$ green cells) in the optic tectum with or without the injury. Very few RG is proliferative in the uninjured DMSO-treated control optic tectum (mean ± SEM; ***p<0.001, ns, p>0.05; two-way ANOVA followed by Tukey's HSD test). See also *Figure 5—source data 1* for quantification. (**K**) Experimental time course of heat shock-induced Notch over-activation experiments shown in (**L–M₃**). Tg(*hsp70l*:gal4 ×*UAS*:NICD-Myc) fish are injured in the optic tecta and followed by heat shocks for three consecutive days (1 hr per day) and are analyzed at 3 dpi. (**L–M₃**) NICD-overexpressed RG (open white arrowheads, red cells) underneath the injury site are not proliferative after the stab injury whereas RG (white arrowheads, green cells) become proliferative in the control optic tectum with the injury. The expression of NICD-Myc is controlled by the gal4-UAS system. It is a mosaic labeling genetic system so that only a subset of cells could be induced to express NICD-Myc. To avoid obscure the signal, only two representative cells were indicated by arrowheads in (**L–M₃**). See also *Figure 5—figure supplement 1C-C₃*. (**N**) Quantification of NICD-Myc$^+$/PCNA$^-$ and NICD-Myc$^+$/PCNA$^+$ RG in (**L–M₃**) showing ~94% NICD-Myc-overexpressed RG are PCNA$^-$. (31/33 cells in 15 sections of 6 fish). The numbers above the bars indicate the animals used. White dashed lines represent the tectal ventricle boundary. RG, radial glia; TeO, tectum opticum; PGZ, periventricular gray zone; TS, torus semicircularis. Scale bars, 100 μm (**B–E**); 50 μm (**F–I**); and 30 μm (**L–M₃**).

DOI: https://doi.org/10.7554/eLife.48660.014

The following source data and figure supplement are available for figure 5:

**Source data 1.** Quantification of PCNA+ cells.

DOI: https://doi.org/10.7554/eLife.48660.016

**Figure supplement 1.** Notch signaling regulates the proliferation of RG.

*Figure 5 continued on next page*

*Figure 5 continued*

DOI: https://doi.org/10.7554/eLife.48660.015

Furthermore, we took advantage of Tg(*hsp70l*:gal4×*UAS*:NICD-Myc) double-transgenic fish, in which a heat shock promoter drives mosaic expression of the NICD-Myc fusion protein, allowing conditional and potent over-activation of Notch signaling (*Figure 5K*) (*Scheer et al., 2001*). Mis-expression of NICD significantly blocked the cell-cycle entry of tectal RG following stab injury, that is, ~94% (31/33 cells, n = 15 sections) of NICD-overexpressed RG underneath the injury sites were PCNA$^-$ (*Figure 5L–N* and *Figure 5—figure supplement 1E-E$_3$*). Torus semicircularis (TS) is the midbrain tissue under the PGZ of the central optic tectum, and their boundary could be unambiguously defined by DAPI staining (*Figure 1—figure supplement 1A-C$_2$* and *Figure 5—figure supplement 1F*-G$_3$). We noticed that stab injury induced some cells in the TS underneath the injury site (close to the boundary of TS and PGZ) to become proliferative in some animals, which required further investigation (*Figure 5L–L$_3$*). In sum, Notch inhibition mediates the stochastic cell-cycle entry of reactive RG after the injury.

## Long-term tracing reveals proliferative RG are gliogenic

To examine the fate outputs of proliferative RG after the injury, we utilized the Cre-loxP system to perform the clonal analysis of single RG after stab injury and analyze clonal cell-type compositions by immunostaining of BLBP, a putative maker for RG, and HuC/D, a putative marker for neurons. Notably, the newborn cells were largely BLBP positive, indicative of RG identity (*Figure 6A–B$_3$*). These results raised an immediate question as to whether injury-induced proliferative RG are gliogenic.

To examine this at the population level, we injected wild-type fish with EdU for six consecutive days after the injury and analyzed EdU$^+$ cells underneath the injury sites at 7 dpi combined with immunostaining for BLBP and HuC/D (*Figure 6C*). The results showed a significant increase of newborn cells (EdU$^+$; uninjured: 7.6 ± 2.1 cells, n = 7; injured: 86.6 ± 6.5 cells, n = 8; mean ± SEM; ***p<0.001; *Figure 6D–H*). Notably, EdU$^+$ newborn cells were largely EdU$^+$/BLBP$^+$ RG (78.6 ± 5.9 cells, 90.5 ± 1.4% of total EdU$^+$ cells, n = 8, mean ± SEM) rather than EdU$^+$/HuC/D$^+$ newborn neurons (3 ± 1.0 cells, 3.3 ± 0.9% of total EdU$^+$ cells, n = 7, mean ± SEM) in the injured hemisphere, indicating that tectal RG largely undergo gliogenesis (*Figure 6H*). As a consequence, glial bulges formed underneath the injury sites (*Figure 6D–E$_3$*). More importantly, when we analyzed EdU$^+$ newborn cells at ~300 dpi (*Figure 6I*), the glial bulges remained and were still largely composed of EdU$^+$/BLBP$^+$ RG (111.3 ± 9.4 cells, 93.6 ± 0.7% of total EdU$^+$ cells, n = 3, mean ± SEM; *Figure 6J–N*). Only a few EdU$^+$ cells were EdU$^+$/HuC/D$^+$ neurons (2 ± 0.6 cells, 1.6 ± 0.4% of total EdU$^+$ cells, n = 3; mean ± SEM, *Figure 6J–N*). EdU$^+$/HuC/D$^+$ neurons were found both in the deep and upper regions of the injured optic tectum (*Figure 6J–K$_3$* and *Figure 6—figure supplement 1A-B$_3$*). Together, stab injury triggers the gliogenesis of tectal RG, resulting in the formation of glial bulges in the zebrafish optic tectum.

After the injury, we often observed a physical wound at the injury site on the surface of the optic tectum (1343 ± 315.7 μm$^2$, n = 10, mean ± SEM; *Figure 6—figure supplement 1C–E*). More strikingly, these stab wounds remained up to 300–400 dpi (1339 ± 768.6 μm$^2$, n = 7, mean ± SEM; p>0.05; *Figure 6—figure supplement 1E and F-I$_3$*). These wounds were surrounded by BLBP signals but without cell nuclei, suggesting that the hypertrophic processes of RG formed a glial scar-like structure surrounding the wound, and thereby blocking the repair of the wound (*Figure 6—figure supplement 1A-B$_3$ and F-I$_3$*). Our results suggest a limited regenerative capacity of the adult zebrafish optic tectum.

## Post-injury Notch inhibition promoted the neurogenesis of reactive RG

Down-regulation of Notch signaling is profoundly implicated in the production of neurons during embryonic CNS development (*Beatus and Lendahl, 1998*; *Artavanis-Tsakonas et al., 1999*). We, therefore, wondered whether Notch inhibition could promote the neurogenesis of proliferative RG. As the number of proliferative RG peaked at 3–4 dpi (*Figure 2K* and *Figure 2—figure supplement 1C*), we examined the fate outputs of tectal RG labeled by EdU at 1–6 dpi with Notch inhibition by

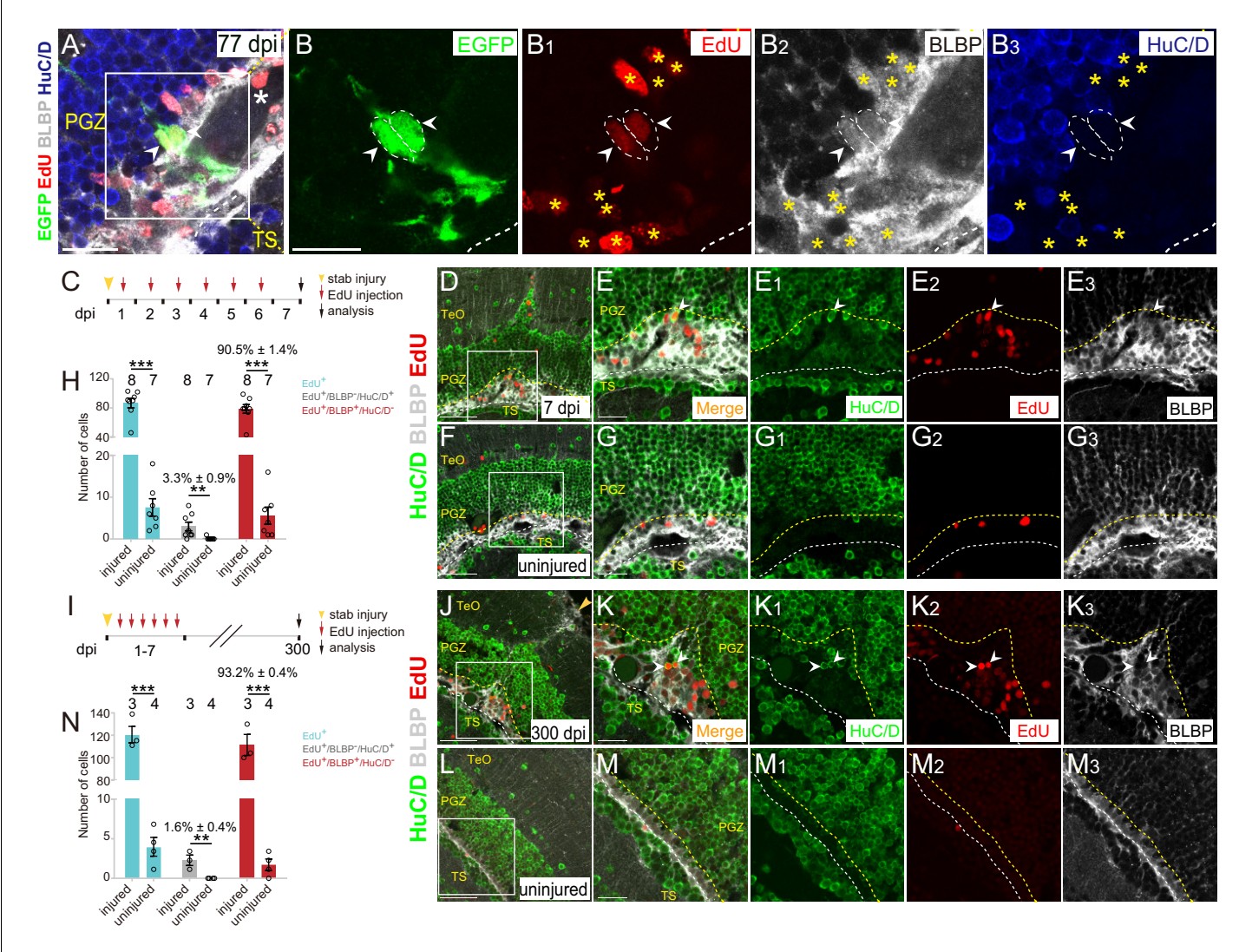

**Figure 6.** Injury-induced RG are largely undergoing gliogenesis. (A–B3) Images of a representative 2 cells clone at 77 dpi. Both cells are EdU+/BLBP+/HuC/D− RG (white arrowheads, white dashed circles). The white asterisk in (A) indicates the blood vessel. Yellow asterisks in (B1–B3) indicate other EdU+/BLBP+/HuC/D− RG underneath the injury site at 77 dpi. (B–B3) The high-magnification images of the boxed area in (A) (C) A schematic showing the procedures used for 7 days EdU pulse-and-stain assay. Fish are injected with EdU for six consecutive days after the injury, the injured and uninjured optic tecta are analyzed at 7 dpi. (D–G3) Representative images of EdU (red), BLBP (gray) and HuC/D (green) immunofluorescences showing that most of the newborn cells are EdU+/BLBP+ RG, while a few newborn cells are EdU+/HuC/D+ neurons (white arrowheads). The newborn RG forms a bulge underneath the injury site. (E–E3) The high-magnification images of the boxed area in (D). (F) The representative image of uninjured optic tectum. (G–G3) The high-magnification images of the boxed area in (F). (H) Quantification of EdU+ newborn cells, EdU+/BLBP+/HuC/D− newborn RG and EdU+/BLBP−/HuC/D+ newborn neurons at 7 dpi. The number of EdU+/BLBP−/HuC/D+ newborn neurons on the injured side is significantly increased compared with that on the uninjured side (mean ± SEM; ***p<0.001, **p<0.01, Wilcoxon test). See also *Figure 6—source data 1* for quantification. (I) A schematic showing the procedure of EdU pulse-and-staining assay for 300 days long-term tracing. Fish are injected with EdU for six consecutive days after the injury, the injured and uninjured optic tecta are analyzed at 300 dpi. (J–M3) Representative images of EdU (red), BLBP (gray) and HuC/D (green) immunofluorescences showing that EdU+ newborn cells that survive up to 300 dpi are largely EdU+/BLBP+ newborn RG, but a few cells are EdU+/HuC/D+ newborn neurons (white arrowheads). (K–K3) The high-magnification representative images of the boxed area in (J). (L) The representative image of uninjured optic tectum. (M–M3) The high-magnification representative images of the boxed area in (L). (N) Quantification of EdU+ newborn cells, EdU+/BLBP+/HuC/D− newborn RG and EdU+/BLBP−/HuC/D+ newborn neurons at 300 dpi (mean ± SEM, ***p<0.001, *p<0.05, Wilcoxon test). See also *Figure 6—source data 2* for quantification. The numbers above the bars indicate the animals used. White dashed lines represent the tectal ventricle boundary. Yellow dashed lines indicate the boundary of bulges. RG, radial glia; TeO, tectum opticum; PGZ, periventricular gray zone; TS, torus semicircularis. Scale bars, 20 μm (A-B3, E-E3, G-G3, K-K3, and M-M3); 50 μm (D, F, J and L).

DOI: https://doi.org/10.7554/eLife.48660.017

The following source data and figure supplement are available for figure 6:

*Figure 6 continued on next page*

*Figure 6 continued*

**Source data 1.** Quantification of EdU+, EdU+/BLBP-/HuC/D+ and EdU+/BLBP+/HuC/D- cells at 7 dpi.
DOI: https://doi.org/10.7554/eLife.48660.019
**Source data 2.** Quantification of EdU+, EdU+/BLBP-/HuC/D+ and EdU+/BLBP+/HuC/D- cells at 300 dpi.
DOI: https://doi.org/10.7554/eLife.48660.020
**Figure supplement 1.** The injury wounds are failed to be restored.
DOI: https://doi.org/10.7554/eLife.48660.018

LY411575 during either 1–3 dpi or 4–5 dpi (*Figure 7A*). Interestingly, although Notch inhibition during both time windows significantly increased the number of newborn cells (EdU$^+$) compared to the control with DMSO treatment during 1–6 dpi (DMSO (1–6 dpi): 75.3 ± 6.6 cells, n = 6; LY411575 (1–3 dpi): 215.4 ± 17.2 cells, n = 7, \*\*\*p<0.001; LY411575 (4–5 dpi): 127.7 ± 12.4 cells, n = 9, \*p<0.05; mean ± SEM; *Figure 7B–E*), newborn neurons (EdU$^+$/HuC/D$^+$) dramatically increased however only in the group in which Notch was inhibited during 4–5 dpi ((DMSO (1–6 dpi): 4.2 ± 1.2 cells, n = 6; LY411575 (1–3 dpi): 6.4 ± 2.1 cells, n = 7, p>0.05; LY411575 (4–5 dpi): 14.2 ± 2.1 cells, n = 9, \*\*p<0.01; mean ± SEM; *Figure 7B–D and F*). The proportion of newborn neurons increased from 5.3 ± 1.1% (DMSO (1–6 dpi), n = 6, mean ± SEM) to 11.9 ± 1.7% (LY411575 (4–5 dpi), n = 9, mean ± SEM; \*p<0.05) (*Figure 7G*). Note that those over-produced neurons always existed as cell clusters, suggesting that they might be clonally related (*Figure 7—figure supplement 1A–F*).

To look into this increased neurogenesis, we further examined fate outputs of tectal RG, which were labeled by EdU during either 1–3 dpi (*Figure 7H*) or 4–6 dpi (*Figure 7I*), with the same treatment of LY411575 (4–5 dpi). The control groups were treated with 0.1% DMSO (4–5 dpi). Fish were sacrificed and cryosections were obtained to stain for EdU and neuronal marker HuC/D at 7 dpi. We found that newborn cells (EdU$^+$; DMSO-treated: 58.4 ± 5.3 cells, n = 24; LY411575-treated: 53.0 ± 3.2, n = 35 cells; mean ± SEM; p>0.05), newborn neurons (EdU$^+$/HuC/D$^+$; DMSO-treated: 3.4 ± 0.3 cells, n = 24; LY411575-treated: 3 ± 0.5, n = 35; mean ± SEM; p>0.05) derived from the proliferative RG labeled during 1–3 dpi with or without Notch inhibition (*Figure 7H*) were indistinguishable in terms of cell number (*Figure 7J, K, N and O*). Moreover, the proportion of newborn neurons showed no significant difference in Notch inhibited fish compared to control fish (DMSO-treated: 6.4 ± 0.6%, n = 24; LY411575-treated: 5.4 ± 0.9%, n = 35; mean ± SEM; p>0.05), which suggested that Notch inhibition itself did not promote the neurogenesis of injury-induced proliferative RG (*Figure 7P*). In contrast, the RG labeled during 4–6 dpi with Notch inhibition (4–5 dpi) (*Figure 7I*) gave rise to much more newborn cells (EdU$^+$) than the control (DMSO-treated: 52.5 ± 6.6 cells, n = 12; LY411575-treated: 107 ± 10.9 cells, n = 18; mean ± SEM; \*\*\*p<0.001; *Figure 7L, M and N*), which was likely due to the cell-cycle entry of reactive RG by Notch inhibition during 4–5 dpi. Interestingly, these increased newborn cells derived from RG labeled during 4–6 dpi with Notch inhibition had a much higher proportion of neurons (DMSO-treated: 4.2 ± 1.2 cells, 6.8 ± 1.9%, n = 12; LY411575-treated: 22.1 ± 3.5 cells, 19.2 ± 2.4%, n = 18; mean ± SEM; \*\*\*p<0.001; *Figure 7L, M, O and P*). Together, our results suggest that post-injury Notch inhibition (4–5 dpi) drives injury-induced reactive RG (non-proliferation) into the cell cycle, producing a significantly higher proportion of neurons compared to those derived from proliferative RG by either the injury (5.3 ± 1.1%, n = 6, mean ± SEM) or Notch inhibition (1.2 ± 0.6%, n = 7, mean ± SEM) (*Figure 7G and Q*).

We further investigated the long-term fate of these newborn neurons. We sacrificed and sectioned the DMSO-treated (4–5 dpi) control fish, which were injected with EdU during 4–6 dpi, at either 7 dpi or 25 dpi (*Figure 7—figure supplement 1G*). The coronal sections were then stained for EdU signals, neuronal marker HuC/D and RG marker BLBP (*Figure 7—figure supplement 1H–K*). The number of newborn cells (EdU$^+$; 7-dpi DMSO-treated: 25.7 ± 2.5 cells, n = 10; 25-dpi DMSO-treated: 41.3 ± 6.1 cells, n = 11; mean ± SEM; \*p<0.05) and newborn RG (EdU$^+$/BLBP$^+$; 7-dpi DMSO-treated: 18.1 ± 2.5 cells, n = 10; 25-dpi DMSO-treated: 33.9 ± 5.8 cells, n = 11; mean ± SEM; \*p<0.05) were increased slightly at 25-dpi compared to 7-dpi DMSO-treated fish (*Figure 7—figure supplement 1H, J, L and M*), which might be due to the variability among different fish. However, the number of newborn neurons (EdU$^+$/HuC/D$^+$; 7-dpi DMSO-treated: 2.4 ± 0.9 cells, n = 10; 25-dpi DMSO-treated: 2.1 ± 0.3 cells, n = 11; mean ± SEM; p>0.05) showed no significant difference (*Figure 7—figure supplement 1N*). Interestingly, the number of newborn neurons in LY411575-treated fish decreased significantly at 25 dpi (EdU$^+$/HuC/D$^+$; 7-dpi LY411575-treated: 15.9 ± 2.7 cells,

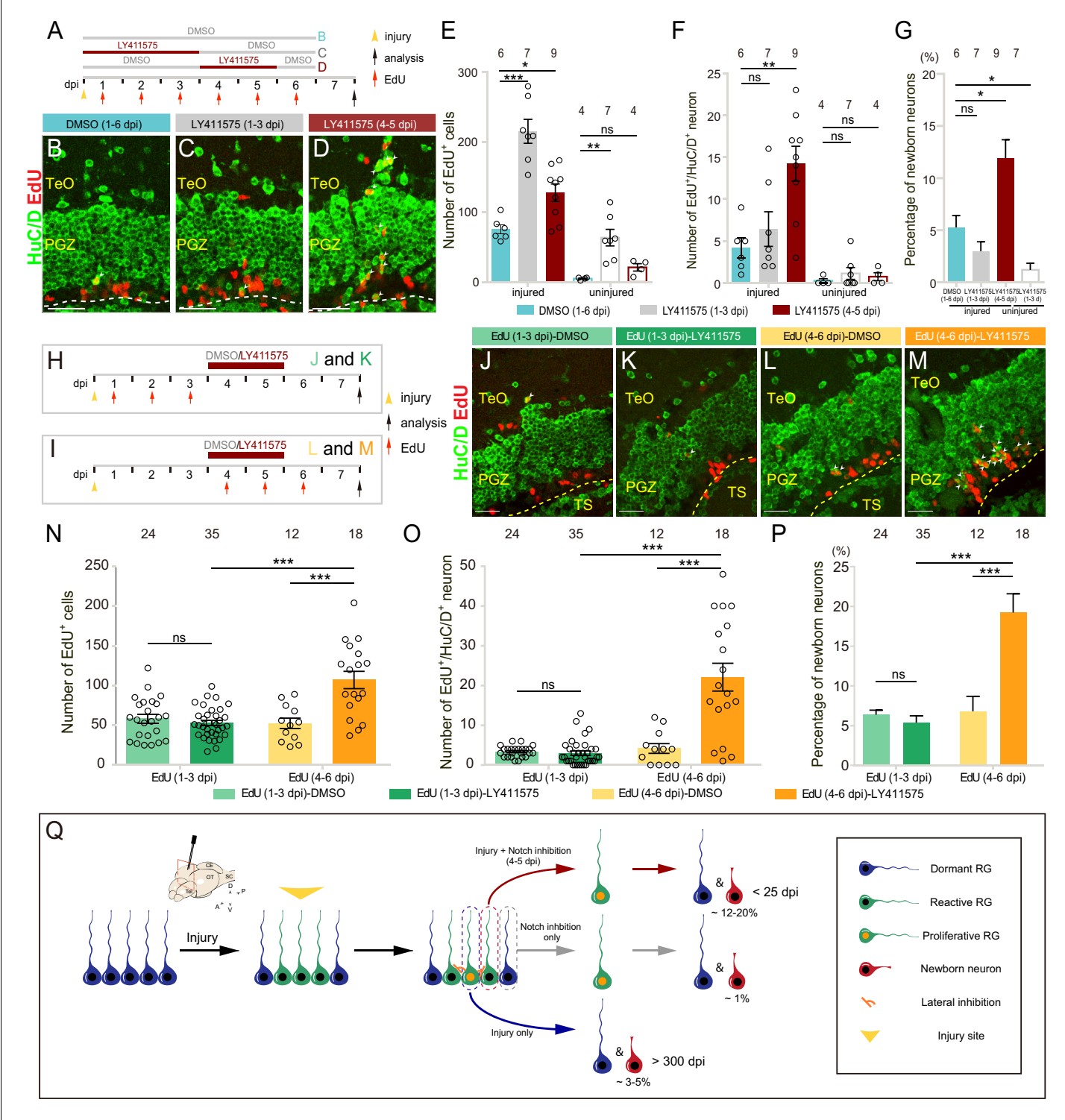

**Figure 7.** Notch inhibition promotes the neurogenesis of reactive tectal RG. (**A**) A schematic of Notch inhibition experiments shown in (**B–D**). In the control group, fish are administrated with DMSO from 1 to 6 dpi. In experimental groups, fish are administrated with LY411575 during either 1 to 3 dpi or 4 to 5 dpi. EdU is injected for six consecutive days after the injury. All the fish are sacrificed and analyzed at 7 dpi. (**B–D**) Representative images of HuC/D (green) and EdU (red) immunofluorescences of 7-dpi optic tecta treated with DMSO for 1–6 dpi, LY411575 for 1–3 dpi, or LY411575 for 4–5 dpi showing that significant more EdU$^+$/HuC/D$^+$ newborn neurons are only generated after the treatment of LY411575 during 4–5 dpi. White arrowheads indicate EdU$^+$/HuC/D$^+$ newborn neurons. (**E–F**) Quantification of EdU$^+$ newborn cells (**E**) and EdU$^+$/HuC/D$^+$ newborn neurons (**F**) in (**B–D**). While Notch inhibition of 1–3 dpi or 4–5 dpi significantly increases the number of EdU$^+$ newborn cells in the injured optic tectum, Notch inhibition during 4–5 dpi

*Figure 7 continued on next page*

*Figure 7 continued*

but not 1–3 dpi significantly increases the number of EdU$^+$/HuC/D$^+$ newborn neurons in the injured optic tectum. In the uninjured optic tecta, Notch inhibition during both 1–3 dpi and 4–5 dpi increases the number of EdU$^+$ newborn cells, but not EdU$^+$/HuC/D$^+$ newborn neurons (mean ± SEM, \*\*\*p<0.001, \*\*p<0.01, \*p<0.05, ns, p>0.05; one-way ANOVA followed by Tukey's HSD test). See also *Figure 7—source datas 1 and 2* for quantification. (G) Proportion of EdU$^+$/HuC/D$^+$ newborn neurons to EdU$^+$ newborn cells in (B–D). Notch inhibition during 4–5 dpi increases the proportion of the neuron production, whereas Notch inhibition during 1–3 dpi decreases the proportion (mean ± SEM, \*\*p<0.01; ns, p>0.05; one-way ANOVA followed by Tukey's HSD test). See also *Figure 7—source data 3* for quantification. (H and I) Schematics of the experimental procedure for Notch inhibition experiments shown in (J-M). After the injury, fish are treated with either DMSO or LY411575 during 4–5 dpi and are injected with EdU for three consecutive days during 1–3 dpi (H) or 4–6 dpi (I). All the fish are sacrificed and analyzed at 7 dpi. (J–M) Representative images of HuC/D (green) and EdU (red) immunofluorescences of the 7-dpi optic tecta after the treatment in (H and I). With the treatment of LY411575 during 4–5 dpi, EdU pluses during 4–6 dpi (L and M) but not 1–3 dpi (J and K) label significant more newborn neurons. White arrowheads indicate EdU$^+$/HuC/D$^+$ newborn neurons. (N and O) Quantification of EdU$^+$ newborn cells (N) and EdU$^+$/HuC/D$^+$ newborn neurons (O) in (J–M) (≥3 replicates for each group; mean ± SEM, \*\*\*p<0.001, ns, p>0.05; two-way ANOVA followed by Tukey's HSD test). See also *Figure 7—source datas 4 and 5* for quantification. (P) Proportion of EdU$^+$/HuC/D$^+$ newborn neurons to EdU$^+$ newborn cells in (J–M). EdU pulses during 4–6 dpi significantly increase the proportion of neuron production (≥3 replicates for each group; mean ± SEM, \*\*\*p<0.001; ns, p>0.05; two-way ANOVA followed by Tukey's HSD test). See also *Figure 7—source data 6* for quantification. (Q) Schematic summary of the working model. Injury induces all RG underneath the injury site to become reactive. Only ~25% of reactive RG enter the cell cycle and become proliferative. The cell-cycle entry of reactive RG is regulated by Notch/Delta lateral inhibition. In the injury condition, proliferative RG largely undergo gliogenesis (~3–5% newborn neurons). The resulting newborn cells could survive up to 300 dpi. In the Notch inhibition condition, dormant RG can become proliferative but only generate ~1% of newborn neurons. However, Notch inhibition during 4–5 dpi drives reactive RG into the cell cycle, giving rise to significant more neurons (~12–20%). Interestingly, these over-produced neurons are largely diminished by 25 dpi. The numbers above the bars indicate the animals used. Yellow dashed lines represent the tectal ventricle boundary. RG, radial glia; TeO, tectum opticum; PGZ, periventricular gray zone; TS, torus semicircularis. Scale bars, 30 μm (B–D); 20 μm (J–M).
DOI: https://doi.org/10.7554/eLife.48660.021

The following source data and figure supplement are available for figure 7:

**Source data 1.** Quantification of EdU+ newborn cells.
DOI: https://doi.org/10.7554/eLife.48660.023
**Source data 2.** Quantification of EdU+/HuC/D+ newborn neurons.
DOI: https://doi.org/10.7554/eLife.48660.024
**Source data 3.** Percentage of EdU+/HuC/D+ newborn neurons.
DOI: https://doi.org/10.7554/eLife.48660.025
**Source data 4.** Quantification of EdU+ newborn cells.
DOI: https://doi.org/10.7554/eLife.48660.026
**Source data 5.** Quantification of EdU+/HuC/D+ newborn neurons.
DOI: https://doi.org/10.7554/eLife.48660.027
**Source data 6.** Percentage of EdU+/HuC/D+ newborn neurons.
DOI: https://doi.org/10.7554/eLife.48660.028
**Figure supplement 1.** Late Notch inhibition-induced over-produced neurons are short-lived.
DOI: https://doi.org/10.7554/eLife.48660.022

n = 10; 25-dpi LY411575-treated: 2.4 ± 0.3 cells, n = 14; mean ± SEM; \*\*\*p<0.001) and became indistinguishable from the DMSO-treated fish (p>0.05; *Figure 7—figure supplement 1I, K and N*). Meanwhile, the number of newborn cells (EdU$^+$; 7-dpi LY411575-treated: 106.9 ± 11.8 cells, n = 10; 25-dpi LY411575-treated: 112.3 ± 11.4 cells, n = 14; mean ± SEM; p>0.05) and newborn RG (EdU$^+$/BLBP$^+$; 7-dpi LY411575-treated: 85.9 ± 9.6 cells, n = 10; 25-dpi LY411575-treated: 99.7 ± 10.7 cells, n = 14; mean ± SEM; p>0.05) showed no significant difference between 7-dpi and 25-dpi LY411575-treated fish (*Figure 7—figure supplement 1L and M*). The remaining newborn neurons could survive until ~86 dpi (*Figure 7—figure supplement 1O–P*). 76% of long-lived neurons resided in the tectum opticum (TeO) in 25-dpi LY411575-treated fish (25 dpi DMSO-treated: 83%; *Figure 7—figure supplement 1Q–S*). We measured the wound area (1669 ± 704.3 μm$^2$, n = 3, mean ± SEM) and found that the post-injury Notch inhibition did not help to complete restoration of the injured optic tectum (*Figure 7—figure supplement 1T and U*).

## Discussion

How do reactive RG enter the cell cycle? Consistent with astrogliosis in injured mammalian CNS, we find that zebrafish tectal RG are undergoing the consecutive phases of glial reactivation and glial proliferation in response to the injury (*Figure 2A–K*) (*Burda and Sofroniew, 2014*). After stab injury,

almost all RG underneath the injury site respond to the injury by glial reactivation, characterized with the up-regulation of vimentin expression and robust GFP expression in a powerful transgenic line Tg (*1016tuba1α*:GFP), which has been used to monitor the injury-induced MG in zebrafish retina (*Fausett and Goldman, 2006*; *Wan et al., 2012*).

After the reactivation, ~25% of RG are entering the cell cycle, and become proliferative (*Figure 2H, L* and *Figure 2—figure supplement 1I-J₂*). More interestingly, we find that the proliferative RG that respond to two sequential injuries are distinct but overlapping, which can be well explained by stochastic cell-cycle entry. It is, unexpectedly, that tectal RG underneath the injury sites randomly enter the cell cycle. How can reactive RG achieve this? Our results show that proliferative RG have high *deltaA* expression while non-proliferating RG have high Notch activity, pointing to the working of Notch/Delta lateral inhibition, reminiscent of the function of Notch/Delta lateral inhibition in stochastically determining the fate of neuroblasts in Drosophila (*Sato et al., 2016*). How is this Notch/Delta lateral inhibition generated after the injury? There are at least two possibilities: 1. The injury induces *deltaA* expression in some RG, leading to a decrease in Notch activity and subsequent cell-cycle entry. As a result of the lateral inhibition, an increased Notch activity keeps neighboring reactive RG in the non-proliferative state. 2. The injury directly blocks Notch activity in some RG that then enter the cell cycle, and leave neighboring cells non-proliferative. It is certainly essential to figure out which model is actually at work.

Can the optic tectum of adult zebrafish regenerate after the injury? Recent studies come to the contradictory answers. Ohshima's lab showed that tectal RG were capable of producing some neurons after the stab injury (~25% of newborn cells) (*Shimizu et al., 2018*), whereas Kaslin's lab reported the opposite result, in which tectal RG only produce glial cells but not neurons (*Lindsey et al., 2019*). Using EdU pulse-and-stain assay and Cre-loxP-based clonal analysis of tectal RG, we unambiguously demonstrate at the single-cell resolution that RG give rise to newborn glial cells (~91% of total newborn cells) with only a few neurons (~3–5% of total newborn cells) after the injury (*Figure 7Q*), which is mostly consistent with the study of Kaslin's lab. In fact, the average number of newborn neurons in the study from Ohshima's lab (~4 BrdU$^+$/HuC/D$^+$ neurons) was mostly similar to what we had (~4 EdU$^+$/HuC/D$^+$ neurons), while the number of total newborn cells in their study (~17 BrdU$^+$ cells) is much smaller than ours (~90 EdU$^+$ cells). Thus, the conclusion of a high proportion of newborn neurons from Ohshima's lab is perhaps due to the underestimated number of total newborn cells. Our scRNA-seq data shows that *hmgb2a* and *hmgb2b* are highly expressed in RG of proliferative state (*Figure 6—figure supplement 1J and K*). *hmgb2* is strongly associated with dormancy/activation transition of adult neural stem cells (NSCs) in mice (*Kimura et al., 2018*), and loss of *hmgb2* compromises gliogenesis and promotes neurogenesis (*Abraham et al., 2013*). In situ hybridization also confirm the expression of *hmgb2a* in the proliferating tectal RG as the result of stab injury (*Figure 6—figure supplement 1L-M₂*), suggesting proliferative tectal RG are likely to be gliogenic. Thus, unlike to RG of other CNS regions in zebrafish (*Dias et al., 2012*; *Goldman, 2014*; *Kizil et al., 2012*; *Kroehne et al., 2011*; *Than-Trong and Bally-Cuif, 2015*), tectal RG resembles mammalian astrocytes in terms of injury response, undergoing the gliogenesis rather than the neurogenesis. Moreover, we also show that newborn glial cells can survive up to ~300 dpi and form a bulge structure of glial cells lining the tectal ventricle, while the wounds at the injury sites remain, which suggests that newborn glial cells are unable to migrate to the injury sites and the wounds are not restored. Notably, we observed the hypertrophic process of RG surrounding the injury sites forms a scar-like structure, which may block the restoration of the wounds (*Figure 6—figure supplement 1B-B₃ and F-G₃*).

What controls fate outputs of proliferative RG? Notch inhibition has been implicated in the production of neurons during embryonic development as well as in various injury contexts (*Dias et al., 2012*; *Louvi and Artavanis-Tsakonas, 2006*; *Wan et al., 2012*). We thus investigated whether Notch inhibition at the proper timing after the injury can switch gliogenesis to neurogenesis. Indeed, we find that post-injury Notch inhibition during 4–5 dpi but not 1–3 dpi results in a significant increase in newborn neurons (*Figure 7A–G*). Our further analysis shows that Notch inhibition (4–5 dpi) of injury-induced reactive RG labeled during 4–6 dpi results in a significant increase of neuron production (*Figure 7I, L, M, O and P*). No significant change of neuron production from injury-induced proliferative RG labeled during 1–3 dpi is observed (*Figure 7H, J, K, O and P*). *Ueda et al.* recently showed that Notch inhibition during 4–7 dpi decreases the number of newborn neurons (*Ueda et al., 2018*), which is in contrast to our findings. This inconsistency is likely to be because the

time window of EdU or BrdU treatment is different. Ueda and colleagues treated the injured fish with BrdU during 2–3 dpi. Instead, we treated the injured fish with EdU during 4–6 dpi in this experiment. This means the RG analyzed from two studies are different. In Ueda's study, BrdU treatment during 2–3 dpi is most likely to label proliferative RG as a result of the cell-cycle entry of dormant RG by stab injury, whereas EdU treatment during 4–6 dpi in our analysis is most likely to label proliferative RG as a result of the cell-cycle entry of injury-induced reactive RG by Notch inhibition instead. Our findings raised an interesting hypothesis, that is, cell states of tectal RG, such as dormant state, reactive state, or proliferative state, influence their fate outcomes in the context of Notch inhibition or injury (*Figure 7Q*). Resolving the mechanistic link of cell state and fate potentials will undoubtedly deepen the understanding of the fate control of injury-reactivated RG. Besides, long-term experiments unexpectedly showed that over-produced neurons disappeared by 25 dpi. Earlier studies also showed that manipulation of Notch signaling could drive more RG into the cell cycle in the injury context (*Dias et al., 2012*; *Ueda et al., 2018*; *Wan et al., 2012*). However, none of them investigated the long-term survival of those newborn cells. Our observation of rapid loss of newborn cells might be due to their intrinsic property of short life or due to the lack of significant neuron loss in stab injured optic tectum. The underlying mechanisms that control the death of over-produced neurons are also appealing to be further investigated.

# Materials and methods

### Key resources table

| Reagent type (species) or resource | Designation | Source or reference | Identifiers | Additional information |
|---|---|---|---|---|
| Strain, strain background (*Danio rerio*) | *gfap*:EGFP | *Bernardos and Raymond, 2006* | ZDB-FISH-150901–29307 | Tg(*gfap*:EGFP)mi2001 |
| Strain, strain background (*Danio rerio*) | *her4.1*:dRFP | *Yeo et al., 2007* | ZDB-TGCONSTRCT-070612–2 | Tg(*her4.1*:dRFP) |
| Strain, strain background (*Danio rerio*) | *1016tuba1α*:GFP | PMID: 16763038 | ZDB-GENO-070321–4 | Tg(*1016tuba1α*:GFP) |
| Strain, strain background (*Danio rerio*) | *olig2*:GFP | *Shin et al., 2003* | ZDB-ALT-041129–8 | Tg(*olig2*:GFP) |
| Strain, strain background (*Danio rerio*) | *mpeg1*:GFP | *Ellett et al., 2011* | ZDB-TGCONSTRCT-170801–5 | Tg(*mpeg1*:GFP) |
| Strain, strain background (*Danio rerio*) | *hsp70l*:gal4 | *Scheer et al., 2001* | ZDB-TGCONSTRCT-070117–42 | Tg(*hsp70l*:gal4) |
| Strain, strain background (*Danio rerio*) | *UAS*:NICD-Myc | *Scheer et al., 2001* | ZDB-TGCONSTRCT-070117–24 | Tg(*UAS*:NICD-Myc) |
| Strain, strain background (*Danio rerio*) | *Tp1bglob*:EGFP | This paper | | Tg(*Tp1bglob*:EGFP) is generated using the plasmid from Dr. Nathan Lawson |
| Strain, strain background (*Danio rerio*) | *her4.1*:mCherryT2ACreER^T2 | This paper | | Tg(*her4.1*:mCherryT2ACreERT2) is generated using the plasmid from Dr. Micheal Brand |
| Strain, strain background (*Danio rerio*) | *hsp70l*:DsRed2(*floxed*)EGFP | This paper | | Tg(*her4.1*:mCherryT2ACreERT2) is generated using the plasmid from Dr. Micheal Brand |
| Antibody | Mouse monoclonal anti-PCNA | Abcam | Cat#Ab29 RRID:AB_303394 | 1:1000 |
| Antibody | Rabbit polyclonal anti-GFPtag | Abcam | Cat# ab13970 RRID:AB_300798 | 1:500 |
| Antibody | Chicken monoclonal anti-GFP | Proteintech Group | Cat#50430–2-AP RRID:AB_11042881 | 1:2000 |
| Antibody | Rabbit polyclonal anti-DsRed2 | Takara Bio | Cat#632496 RRID:AB_10013483 | 1:1000 |
| Antibody | Mouse monoclonal anti-HuC/D | Invitrogen | Cat#A21271 RRID:AB_221448 | 1:1000 |

*Continued on next page*

*Continued*

| Reagent type (species) or resource | Designation | Source or reference | Identifiers | Additional information |
|---|---|---|---|---|
| Antibody | Rabbit polyclonal anti-BLBP | Abcam | Cat#ab32423 RRID:AB_880078 | 1:1000 |
| Antibody | Rat monoclonal anti-BrdU | Abcam | Cat#Ab6326 AB_305426 | 1:1000 |
| Recombinant DNA reagent | pTol2-*Tp1bglob*:EGFP | *Quillien et al., 2014* | Addgene plasmid #73586 RRID:Addgene_73586 | Dr. Nathan Lawson (UMass Medical School, Worcester, USA) |
| Recombinant DNA reagent | pTol2-*her4.1*:mCherryT2ACreER^T2 | *Kroehne et al., 2011* | | Dr. Michael Brand (Technische Universität Dresden, Dresden, Germany) |
| Recombinant DNA reagent | pTol2-*hsp70l*:DsRed2(*floxed*)EGFP | *Kroehne et al., 2011* | | Dr. Michael Brand (Technische Universität Dresden, Dresden, ermany) |
| Recombinant DNA reagent | pBS:*dlA* clone 4 | PMID: 9425133 | | Dr. Judith S. Eisen (University of Oregon, Eugene, USA) |
| Commercial assay or kit | Single Cell 3' Library and Gel Bead kit v2 Chip kit | 10x Genomics | 120237 | |
| Commercial assay or kit | dsDNA High Sensitivity Assay Kit | AATI | DNF-474–0500 | |
| Commercial assay or kit | High Sensitivity Large Fragment −50 kb Analysis Kit | AATI | DNF-464 | |
| Commercial assay or kit | MEGAscriptTM T7 High Yield Transcription Kit | Invitrogen | AM1334 | |
| Commercial assay or kit | Click-iT EdU imaging kit | Invitrogen | C10340 | |
| Commercial assay or kit | DIG RNA labeling kit | Roche | 11277073910 | |
| Chemical compound, drug | Papain | Worthington Biochemical Corporation | LS003126 | 100 μl in 5 ml DEME/F12 |
| Chemical compound, drug | Tamoxifen | Sigma | T5648 | 2.5–5 μM |
| Chemical compound, drug | LY411575 | Selleck Chemical | S2714 | 10 μM |
| Software, algorithm | Cell Ranger Single Cell Software Suite (v2.1.0) | 10x genomics | https://support.10x genomics.com | |
| Software, algorithm | Seurat R package | Satjia lab | http://satijalab.org/seurat/ | |
| Software, algorithm | R 3.5.1 | R-project | https://www.r-project.org/ | |
| Software, algorithm | GraphPad Prism | GraphPad Software | www.graphpad.com | |
| Software, algorithm | FIJI | PMID: 22743772 | http://fiji.sc/ | |
| Software, algorithm | FV10-ASW 4.0 Viewer | Olympus | www.olympus-global.com | |

## Zebrafish husbandry and transgenic lines

Zebrafish embryos, larvae, and adults were produced, grown, and maintained at 28°C according to standard protocols except during heat shock treatments. Embryos were harvested and kept in the embryo medium (0.294 g/L NaCl, 0.0127 g/L KCl, 0.0485 g/L CaCl$_2$·2H$_2$O, 0.0813 g/L MgSO4·7H$_2$O, 0.3 g/L sea salt, and $2 \times 10^{-4}$ g/L methylene blue) at 28°C. Young adult zebrafish ranging in age from 2 to 4 months old were used for experiments. Approximately equal sex ratios were used for experiments. All young adult fish were fed twice daily.

Published lines used in this study include: Wild type, Tg(*gfap*:EGFP)[mi2001] (ZDB-FISH-150901–29307) (*Bernardos and Raymond, 2006*), Tg(*her4.1*:dRFP) (ZDB-TGCONSTRCT-070612–2) (*Yeo et al., 2007*), Tg(*1016tuba1α*:GFP) (ZDB-GENO-070321–4) (*Fausett and Goldman, 2006*), Tg (*olig2*:GFP) (ZDB-ALT-041129–8) (*Shin et al., 2003*), Tg(*mpeg1*:GFP) ZDB-TGCONSTRCT-170801–5

(*Ellett et al., 2011*), Tg(*hsp70l*:gal4) (ZDB-TGCONSTRCT-070117–42) (*Scheer et al., 2001*), Tg(*UAS*:NICD-Myc) (ZDB-TGCONSTRCT-070117–24) (*Scheer et al., 2001*). Details of the generation of the new lines generated in this study are described below. At least three independent founders of each line were screened and checked to confirm the described expression patterns.

### Generation of Tg (*Tp1bglob*:EGFP) line

The plasmid of pTol2-*tp1bglob*:EGFP (Addgene plasmid # 73586) was a gift from Dr. Nathan Lawson (UMass Medical School, Worcester, USA) (*Quillien et al., 2014*). The plasmid was co-injected with *Tol2* mRNA at one-cell stage. Zebrafish embryos were grown and maintained according to standard protocols. In this line, cells with a high Notch signaling level express the fluorescent protein EGFP. The full name of this line is Tg(*Tp1bglob*:EGFP).

### Generation of Tg(*her4.1*:mCherryT2ACreER^T2^) line

The plasmid was a gift from Dr. Michael Brand (Technische Universität Dresden, Dresden, Germany) (*Kroehne et al., 2011*). The plasmid was co-injected with *Tol2* mRNA at one-cell stage. Zebrafish embryos were grown and maintained according to standard protocols. In this line, radial glia express the fluorescent protein mCherry followed by a CreER$^{T2}$ element. The full name of this line is Tg(*her4.1*:mCherryT2ACreER$^{T2}$).

### Generation of Tg(*hsp70l*:DsRed2(*floxed*)EGFP) line

The plasmid was a gift from Dr. Michael Brand (Technische Universität Dresden, Dresden, Germany) (*Kroehne et al., 2011*). The plasmid was co-injected with *Tol2* mRNA at one-cell stage. Zebrafish embryos were grown and maintained according to standard protocols. In this line, cells express DsRed2 only after heat shock. Upon CreER$^{T2}$ mediated recombination, the DsRed2-floxed cassette is eliminated and cells exposed to heat shock then express the fluorescent protein EGFP. The full name of this line is Tg(*hsp70l*:DsRed2(*floxed*)EGFP).

### Stab injury of the optic tectum of adult zebrafish

Fish were anesthetized using 0.02% MS-222 for 30 to 45 s (s). Fish were placed in a piece of 0.02% MS-222 soaked tissue, and a set of tweezers was used to place them properly, allowing accessibility to the head. With the visual aid of a dissecting microscope, the needle (30 gauge, outer diameter 300 µm) was stabbed ~400 µm deep into the optic tectum through the skull. After the injury, fish were returned back to the fish tank.

### Single-cell sample preparation

The single-cell suspension of adult zebrafish optic tecta was prepared by following a published protocol (*Lopez-Ramirez et al., 2016*). Large-area injuries were introduced to the central-dorsal part of the optic tecta of ~2 months old Tg(*gfap*:EGFP) fish and the fish were returned back to a fish tank under standard conditions. At three dpi, the fish were anesthetized and sacrificed. The optic tecta were dissected and dissociated by the digestion in 350 µl papain solution at 37°C for 15 min (mins). During digestion, the tissues were pipetted up and down 4 × 10 times. Digestion was stopped by 1400 µl washing buffer. The cell solution was filtered with a 40 µm cell strainer (BD Falcon), and then centrifuged at 200 g for 5 mins at 4°C. The supernatant was discarded and the pellets were resuspended with 1 × PBS with 0.04% BSA. Then fluorescence-activated cell sorting (FACS) was performed, collecting the cells with high EGFP fluorescence into 1 × PBS with 0.04% BSA.

Papain solution: To prepare the papain solution, add 100 µl papain (Worthington, LS003126), 100 µl DNase (1%, Sigma, DN25) and 200 µl L-cysteine (12 mg/ml, Sigma, C6852) into 5 ml DMEM/F12 (Invitrogen, 11330032).

Washing solution: To prepare the washing solution, add 65 µl glucose 45% (Invitrogen, 04196545 SB), 50 µl HEPES 1M (Sigma, H4034) and 500 µl FBS (Gibco, 10270106) in 9.385 ml DPBS 1× (Invitrogen, 14190–144). All solutions were filtered through a 0.22 µm pore size filter (Millipore) to sterilize and stored at 4°C before use.

## Single-cell RNA sequencing

To perform single-cell RNA sequencing (scRNA-seq), cells after FACS were loaded onto the Chromium Single Cell Chip (10x Genomics, USA) according to the manufacturer's protocol. The scRNA-seq libraries were generated using the GemCode Single-Cell Instrument and Single Cell 3' Library and Gel Bead kit v2 Chip kit (10x Genomics, 120237) by following the manufacturer's protocol. Library quantification and quality assessments were performed by Qubit fluorometric assay (Invitrogen) with dsDNA High Sensitivity Assay Kit (AATI, DNF-474–0500) and the fragment analyzer with High Sensitivity Large Fragment −50 kb Analysis Kit (AATI, DNF-464). The indexed library was tested for quality, and sequenced by the Illumina NovaSeq 6000 sequencer with the S2 flow cell using paired-end 150 × 150 base pair as the sequencing mode. The sequencing depth was 60K reads per cell.

## Single-cell sequencing data analysis

Single-cell FASTQ sequencing reads (Novogene) were processed, and converted to digital gene expression matrices after mapping to the zebrafish genome (Zv10) using the Cell Ranger Single Cell Software Suite (v2.1.0) provided on 10x genomics website (https://support.10xgenomics.com/single-cell-gene-expression/software/pipelines/ latest/what-is-cell-ranger). 66,817 mean reads per cell and 1325 mean genes per cell were obtained.

For further analysis, we used an analysis pipeline provided by Seurat R package (http://satijalab.org/seurat/). Firstly, the Seurat object was created to filter low-abundance genes, cell doublets and low-quality libraries (with low gene numbers and high mitochondrial transcripts). Secondly, the filtered data were normalized and used to identify highly variable genes based on expression and dispersion. Thirdly, the data were scaled, and the unwanted sources of variation were removed. Fourthly, cell clustering analyses were performed by the t-SNE projection (*Figure 3—figure supplement 1B*). Finally, we found out the markers for every cluster (*Figure 3—figure supplement 1C*). Due to possible contamination during tissue dissociation and FACS, the samples were contaminated with other types of cells from the optic tecta and other neighboring tissues. Based on the markers of each cluster, these contaminated cells were identified and removed after the initial clustering.

Non-glial cell clusters (1, 2, 5, 6, 11, 12, 13, 14) were identified by high expression of neuronal markers such as *neurod1*, *elavl3*, *gad1b* and *slc17a6b* and low expression of glial markers, dormant and proliferative cell markers such as *fabp7a*, *gfap*, *her4.1*, *mfge8a* and *pcna* (*Figure 3—figure supplement 1D and E*). These clusters were removed and the remaining cells were used for further analysis.

As many proliferative progenitors are present in the tectal proliferation zone (TPZ) (*Galant et al., 2016*; *Ito et al., 2010*), a big-area injury induced a lot of PCNA[+] tectal RG at 3 dpi. We obtained two *pcna*[+] clusters in the t-SNE plot (cluster 1 and 2, *Figure 3—figure supplement 2A and B*). However, based on experimental evidence: 1. Injury caused the obvious down-regulation of *her4* PCNA[+] RG, whereas *her4* was highly expressed in RG in TPZ (*Figure 3—figure supplement 2B–F*); 2. The previous study showed progenitors in TPZ were able to generate oligodendrocytes. We did not find any new-born cell derived from injury-induced PCNA[+] RG was oligodendrocyte, and *olig2* expression was noticed in cluster 1 and 10 but not in cluster 2 (*Figure 3—figure supplement 2B and C*). We identified cluster 1 as the progenitors in the TPZ, and it was removed from our data. Following these step-wise filtering processes, we obtained the purified data of each sample.

## Cell cycle phase analysis

To obtain the cell-cycle properties of the cells in our sample, the 'CellCycleScoring' function of Seurat was used. Briefly, each cell was scored based on its expression of G2/M and S phase marker genes. Then the numbers of cells in different cell cycle phases were counted and the ratios of individual cell cycle phases were calculated.

## Gene-gene correlation analysis

The gene-gene correlation was measured according to the pairwise Pearson correlational distances. 'bioDist' R package was used to calculate these correlational distances.

### Pseudo-time trajectory analysis

After the t-SNE cluster analysis of the single-cell data, trajectory analysis was performed to investigate the pseudo-time of four identified states by using 'monocle' and 'Slingshot' R package.

### Pharmacological inhibitors treatments

LY411575 (final concentration of 10 µM, Selleck Chemical, S2714), RO4929097 (final concentration of 50 µM, Selleck Chemicals, S1757) or DMSO (Dimethyl sulfoxide, final concentration of 10 µM, Sigma, B8418) was applied freshly to the fish water at 28°C in the dark for desired days (18 hr per day). The LY411575, RO4929097, or DMSO solutions were changed twice a day.

### Heat shock-induced Notch over-activation

Double transgenic fish Tg(hsp70l:gal4 ×UAS:NICD-Myc) or wild-type fish were heat-shocked in a warm water bath at 38°C for 1 hr on three consecutive days and retrieved to their tank at 28°C. As the mosaic expression of the transgene, the same sections contained NICD-Myc-overexpressing and control cells. Myc expression was detected after heat shock.

### Tamoxifen and heat shock treatments

To induce CreER$^{T2}$ mediated recombination, tamoxifen (TAM, final concentration of 2.5–5 µM, Sigma, T5648) was applied to the fish water at 28°C in the dark for three days (12 hr per day). Double transgenic fish Tg(her4.1:mCherryT2ACreER$^{T2}$::hsp70l:DsRed2(floxed)EGFP) were heat-shocked at 38°C for 1 hr once daily on three consecutive days before sacrifice. 6 hr after the last heat shock, fish were sacrificed for analysis.

### BrdU labeling

Bromodeoxyuridine (BrdU, final concentration of 10 mM, Sigma, B5002) was applied freshly to the fish water at 28°C in the dark for desired days (12 hr per day).

### EdU labeling and detection

Zebrafish were anesthetized, placed on a wet tissue, and injected intraperitoneally with ~5 µl 5 mM 5-ethynyl-2′-deoxyuridine (EdU) in 0.1 M sterile PBS. After injection, fish were retrieved to a fish tank and used for further experiments. To detect the EdU signal, the EdU Click-iT reaction solution (Invitrogen, C10340) was prepared freshly according to the manufacturer's protocol. Sections on slides were covered with a solution and incubated in a humid chamber at room temperature in the dark for 1 hr. After three 10 mins wash in PBS, sections were used for imaging or subsequent processing for immunohistochemistry.

### Tissue preparation and immunohistochemistry

Brains were fixed in 4% paraformaldehyde (PFA, Electron Microscopy Services, USA, 157–8) overnight, cryoprotected in 30% sucrose for 6 hr, flash-frozen and cryosectioned at a thickness of 12 µm. The fluorescent immunochemistry was performed on brain sections as described (*Tang et al., 2017*). Sections were washed with 1 × PBS for 10 mins for three times and permeabilized in 1 × PBS with 0.5% Triton X-100 for 30 mins. After blocking with 5% BSA solution (Sigma) at RT for 1 hr, sections were incubated with the primary antibody at 4°C overnight. Sections were then washed with 1 × PBS and incubated with Alexa Fluor 488-, 594-, or 647-conjugated secondary antibody (1:1,000; Jackson Immuno Research Laboratories Inc) at room temperature for 2 hr. 4′,6-diamidino-2-phenylindole (DAPI) staining was performed according to the standard protocol. Slides were finally mounted using the fluorescent mounting medium (Sigma). For PCNA, HuC/D staining, sections were pre-treated with Improved Citrate Antigen Retrieval Solution (Beyotime Biotechnology, P0090) for 5 mins and washed by Washing Buffer (Beyotime Biotechnology, P0106C) for twice and 1 × PBS for once before blocking or in the citrate acid buffer (10 mM, 0.05% Tween 20, pH 6.0) at 95°C for 30 mins. For BrdU staining, sections were treated with 2 N HCl at 37°C for 10 mins followed by neutralization with 0.1 M sodium borate solution at room temperature for 10 mins and washed by 1 × PBS for 10 mins for three times. Primary antibodies used in this study are listed in the Key resources table.

## In situ hybridization

The digoxigenin (DIG)-labeled *her4*, *mfge8a*, *dla*, *klf6a*, *insm1a* and *hmgb2a* antisense probes were prepared by using the MEGAscriptTM T7 High Yield Transcription Kit (Invitrogen, AM1334) and DIG RNA labeling kit (Roche, 11277073910). The cDNA of each gene was amplified by PCR using the following primers: *her4.1*-F:5'-CCCTCGAGCTGATCCTGACGGAGAACTGAACAC-3'; *her4.1*-R:5'-TAA TACGACTCACTATAGTTCTAGAATAGACGAAGAGAAA ACAAACC-3'; *mfge8a*-F: 5'-TGCAGCC- CAAACCCATGTAA-3'; *mfge8a*-R:5'-TAATACGACTCACTATAGGGTGAGTCGGGATTTCATGCCC- 3'; *klf6a*-F:5'-ATGGATGTTCTACCAATGTGCAGCA-3'; *klf6a*-R:5'-TAATACGACTCACTATAGGG TCAGAGGTGCCTCTTCATGTGC-3'; *insm1a*-F:5'-ATGCCCAGAGGATTTTTAGTCAAGC-3'; *insm1a*- R:5'-TAATACGACTCACTATAGGGTTGTCTTCAGCAGGCTGGAC GC-3'; *hmgb2a* F:5'-ATGGG TAAAGATCCAAATAAGCCCAG-3'; *hmgb2a*-R:5'-TAATACGACTCACTATAGGGTTATTCGTCATCA TCATCCTCGTCCTC-3'.

The injured and uninjured zebrafish brains were fixed in 4% PFA at 4°C overnight followed by dehydration in 30% sucrose and then were cryosectioned at a thickness of 12 μm. The slices were post-fixed in 4% PFA at room temperature for 15 mins and washed with 1 × PBS at room temperature for 3 mins. To block the activity of endogenous peroxidase, all slides were treated with 0.1% $H_2O_2$ at room temperature for 30 mins. After being washed twice with 1 × PBS at room temperature for 3 mins, slides were treated with 10 μg/ml proteinase K (Sigma) diluted in TE (10 mM Tris-HCl, pH 8.0, and 1 mM EDTA, pH 8.0) at 37°C for 8 mins, then treated with 4% PFA at room temperature for 10 mins. Subsequently, all slides were washed with 1 × PBS at RT for 3 mins, followed by the incubation in 0.2 M HCl at RT for 10 mins. After being washed with 1 × PBS for 5 mins, all slices were then incubated with 0.1 M triethanol amine-HCl (662.5 μl triethanolamine and 1.35 ml 1 M HCl; adding water to the final volume of 50 ml, pH 8.0) at room temperature for 1 min and in 0.1 M triethanol amine-HCl containing 0.25% acetic anhydrate at room temperature for 10 mins with gentle shaking. Slides were then washed by 1 × PBS at room temperature for 5 mins, then were dehydrated in a series of 60%, 80%, 95%, and twice in 100% ethanol at room temperature for 90 s, respectively. Slides were incubated in the hybridization buffer (50% formamide (Sigma), 10 mM Tris-HCl, pH 8.0, 200 μg/ml yeast tRNA (Invitrogen), 1 × Denhart buffer, SDS, EDTA and 10% dextran sulfate (Ambion) containing 1 μg/ml probes at 60°C overnight. On the second day, slides were washed sequentially in 5 × SSC at 65°C for 30 mins, 2 × SSC with 50% formamide at 65°C for 30 mins, TNE buffer (100 ml TNE consisting of 1 ml 1 M Tris-HCl, pH 7.5, 10 ml 5 M NaCl, and 0.2 ml 0.5 M EDTA) at 37°C for 10 min and then in TNE buffer with 20 μg/ml RNaseA at 37°C for 30 mins. Slides were then incubated with 2 × SSC at 60°C for 20 mins, 0.2 × SSC at 60°C for 20 mins, and 0.1 × SSC at RT for 20 mins. Next, slides were blocked by TN buffer at room temperature for 5 mins (200 ml TN buffer consisting of 20 ml 1 M Tris-HCl, pH 7.5, 6 ml 5 M NaCl, and 174 ml water) followed by TNB buffer (TN buffer + 0.5% blocking reagent; Roche) at room temperature for 5 mins. Finally, slides were incubated in TNB buffer with anti–DIG-POD (1:500; Roche) at 4°C overnight. On the third day, the signal was detected by the TSATM Plus Cyanine 3/Fluorescein System (PerkinElmer, NEL753001KT).

## Imaging

Images were taken using an inverted confocal microscope system (FV1200, Olympus) confocal microscope using 10 × (air, 0.4 NA), 30 × (silicon oil, 1.05 NA), or 60 × (silicon oil, 1.3 NA) objectives.

## Quantifications and statistical analysis

All quantification and visualization were performed with FV10-ASW 4.0 Viewer (Olympus), and Image J. Adobe Illustrator CS6 was used to process acquired 2D *Figure 3D* image stacks were analyzed using Imaris software (Bitplane). 12-μm-thick sections (around the injury sites, 8–14 sections per tectum) between the anterior optic tectum and the posterior optic tectum were used for statistical analyses. For cell counting, cryosections of the injury sites were analyzed (every second serial section). Microsoft Excel was used to process the measured data.

To perform the statistical analysis, p values were calculated with GraphPad Prism (or Microsoft Excel). The unpaired, non-parametric Wilcoxon test was applied for comparison of two groups. The one-way ANOVA, followed by Tukey's HSD test was applied for comparison of different groups with one treatment. The two-way ANOVA followed by Tukey's HSD test was applied for comparison of

four groups with two treatments. Error bars represent SEM. ****p<0.0001, ***p<0.001; **p<0.01; *p<0.05; ns, p>0.05.

## Acknowledgements

We thank Dr. Patricia Jusuf for her great help on editing the manuscript, Dr. Michael Brand and Dr. Judith S Eisen for the plasmids, Dr. Su Guo, Dr. Jiulin Du, Dr. Weijun Pan, Dr. Hui Xu and Dr. Xu Wang for fish lines, Songlin Qian, Haiyan Wu and Lijuan Quan from the FACS Facility at the Institute of Neuroscience (ION) for assistance with FACS, Dr. Min Zhang and Zhenning Zhou from Molecular and Cellular Biology Core Facility at the ION for assistance with single-cell RNA library construction, Baijie Xu, Xia Tang, Hui Zhang and Lei Du for their assistance with single-cell sequencing data analysis, Mengmeng Jin, Yan Li, Shui Yu, Huiwen Qin, Yuan Fang for their help of plasmids construction, Xinling Jia for in situ hybridization, Xiaoying Qiu for fish care. We are grateful to Dr. Patricia Jusuf, Dr. Daniel Goldman, Dr. Su Guo, Dr. Hui Xu, Dr. Laure Bally-Cuif, and members of He's laboratory for helpful discussion and suggestions. This work was supported by the Shanghai Municipal Science and Technology Major Project (Grant No. 2018SHZDZX05), Strategic Priority Research Program of Chinese Academy of Science (Grant No. XDB32000000), State Key Laboratory of Neuroscience, Shanghai basic research field Project (Grant No.18JC1410100), National Natural Science Foundation of China (Grant No. 31471042), China Thousand Talents Program.

## Additional information

### Funding

| Funder | Grant reference number | Author |
|---|---|---|
| Shanghai Municipal Science and Technology Major Project | 2018SHZDZX05 | Jie He |
| Chinese Academy of Sciences | Strategy Priority Research Program XDB32000000 | Jie He |
| Shanghai Basic Research Field Project | 18JC1410100 | Jie He |
| National Natural Science Foundation of China | 31471042 | Jie He |
| China Thousand Talents Program | | Jie He |

The funders had no role in study design, data collection and interpretation, or the decision to submit the work for publication.

### Author contributions

Shuguang Yu, Conceptualization, Data curation, Software, Formal analysis, Validation, Investigation, Visualization, Methodology, Writing—original draft, Writing—review and editing; Jie He, Conceptualization, Resources, Data curation, Supervision, Funding acquisition, Investigation, Writing—original draft, Project administration, Writing—review and editing

### Author ORCIDs

Shuguang Yu https://orcid.org/0000-0001-6640-5420
Jie He https://orcid.org/0000-0002-2539-2616

### Decision letter and Author response

Decision letter https://doi.org/10.7554/eLife.48660.033
Author response https://doi.org/10.7554/eLife.48660.034

# Additional files

## Supplementary files
• Transparent reporting form
DOI: https://doi.org/10.7554/eLife.48660.029

## Data availability
Data has been deposited in Dryad (https://doi.org/10.5061/dryad.31t3425).

The following dataset was generated:

| Author(s) | Year | Dataset title | Dataset URL | Database and Identifier |
|---|---|---|---|---|
| Shuguang Yu, Jie He | 2019 | Data from: Stochastic cell-cycle entry and cell-state-dependent fate outputs of injury-reactivated tectal radial glia in zebrafish | https://doi.org/10.5061/dryad.31t3425 | Dryad Digital Repository, 10.5061/dryad.31t3425 |

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
