## [Decision Letter]

[Editors’ note: a previous version of this study was rejected after peer review, but the authors submitted for reconsideration. The first decision letter after peer review is shown below.]

Thank you for choosing to send your work, "Notch-dependent stochastic cell-cycle entry of injury-reactivated tectal radial glia", for consideration at *eLife*. Your submission has been assessed by a Senior Editor in consultation with a member of the Board of Reviewing Editors. Although the work is of interest, we regret to inform you that the findings at this stage are too preliminary for further consideration at *eLife*. If at some later stage you feel you can make these revisions, we would be open to receiving a new manuscript that addresses all of the concerns of the reviewers.

Specifically, the following revisions are required and described in further detail in the full reviews below:

• This paper needs to be rewritten.

• New Introduction should be provided with more focus on fish data.

• The Results section, figures and figure legends need to be reorganized and, in some case, extended.

• Stab wound Injuries need to be done at different geographical regions of the optic tectum.

• The capacity of the optic tectum to regenerate or not need to be investigated and discussed.

• They need to show to which structure or tissue the newly generated RG cells or neurons (late treatment of notch) are giving rise.

Please note that we aim to publish articles with a single round of revision that would typically be accomplished within two months. This means that work that has potential, but in our judgment would need extensive additional work, will not be considered for in-depth review. We do not intend any criticism of the quality of the data or the rigor of the science. We wish you good luck with your work and we hope you will consider *eLife* for future submissions.

Reviewer #1:

In this manuscript the authors investigate the molecular mechanisms of radial glia (RG) activation and their cell fate upon optic tectum injury of adult zebrafish. They show that upon stab wound injury RG cells of the periventricular gray zone (PGZ), which are otherwise quiescent, enter the cell-cycle in a stochastic manner. Fate map studies demonstrate that the majority of activated tectal RG cells will undergo gliosis instead of neurogenesis. Using single-cell RNA-seq, the authors identified 4 distinct RG cell states: quiescent, activated, and 2 sub-proliferative phases. Their data suggest that downregulation of notch target genes, such as *her4.1*, controls the transition from the reactive to the proliferative state of the RG cells. Furthermore, gain-of-function (heat-shock induced activation) and loss-of-function (pharmacological inhibition) experiments targeting notch signaling confirmed the role of this pathway in controlling the cell cycle state of RG cells and their cell fate. Pharmacological inhibition of the notch pathway early post-injury leads to gliosis while late notch suppression (after 3 days post injury) leads to neurogenesis.

In principle, this is an interesting topic in a very important field of research.

However, in my opinion, the majority of the conclusions drawn from the presented work are not novel.

For example, the activation of RG cells upon injury and the role of notch in RG cell state and fate have been largely addressed in several earlier papers from the labs of Toshio Ohshima and Laure Bally-Cuif. I particularly invite the authors to refer to the publication by Ueda et al. from 2018, which deals with the role of notch signaling in regenerative neurogenesis in the adult zebrafish optic tectum (please cite this paper in your next submission).

Figures and figure legends are not well organized, which makes it difficult for the reader to follow and understand them, for example:

- Figure 1A consists of 4 panels which are not labelled (the same is true for other figures).

- There are 2 panels labelled 1B but no panel 1C.

- In Figure 1G: where is the cell count for RG cells?

- The figure legend of Figure 1 and in general all other figure legends in this work are very poor and cryptic. The authors need to better describe the figures. The reader needs to be guided through the results and the figures and should not have to guess what the authors want to show.

In Figure 2, at 5 dpi apparently there is an increase in GFP expression and a lot of proliferation close to the site of injury. What kind of cells are these? Please describe and investigate this issue further. At least discuss it in the Discussion section.

For some injury experiments the authors use 2 months old zebrafish. Why?

To my knowledge, zebrafish are considered adults only after 3 months post fertilization.

In their manuscript the authors suggest that injury of the zebrafish adult optic tectum leads to gliogenesis. This result is very interesting, but partially in contradiction with previous results reported by Shimizu et al., 2018, and Ueda et al., 2018, which showed that injury induces proliferation of RG cells and subsequent formation of new neurons. These contradictions need to be discussed by the authors.

Galant et al. showed in 2016 that new neurons are generated only from RG cells of the PGZ, close to the tectal proliferation zone (TPZ) of the optic tectum. This suggests that the region where the injury is inflicted can decide between gliogenesis or neurogenesis.

In their approach the authors observed an increase in gliogenesis but fail to describe and discuss the consequences. Do they see regeneration in the optic tectum? Is there scar formation? These are very important points which also need to be addressed.

Reviewer #2:

In this study the authors report that the proliferation of the radial glia of the zebrafish tectum after stab injury is dependent on Notch. The authors use a combination of immunolabeling and single cell RNA-seq to demonstrate this relationship, and do functional tests with Notch inhibitors and NICD over-expression. Similar results have been obtained in other areas of the nervous system in fish (e.g. retina), and so this result is not really surprising. This is a largely descriptive study, but what is needed is a mechanistic understanding of the connections between Notch signaling and proliferation in the fish glia, and this is lacking.

1) The focus on a "salt-and-pepper" is a bit odd. It looks more like clumps of labeled cells around the site of the stab injury. (e.g. Figure 4D).

2) The use of PCNA is the primary label for proliferating cells, but it is important to note that this is not a reliable indicator of proliferating cells. Additional markers, such as EdU labeling are necessary to substantiate this.

3) In most experiments, small numbers of fish were analyzed with relatively small numbers of cell counted. The results should be substantiated with larger numbers.

4) There are many grammatical errors throughout the manuscript and the authors should have this edited for English.

5) In some figures, the excessive number and location of arrows prevents one from actually seeing the result (Figure 4E).

Reviewer #3:

This manuscript describes an interesting body of work that should be an important contribution of new information to the field of brain injury. The experimental design is logical and rigorous, the data mostly provide strong support for the claims and conclusions made by the authors and the figures are informatively presented.

[Editors’ note: what now follows is the decision letter after the authors submitted for further consideration.]

Thank you for submitting your article "Notch inhibition mediates stochastic cell-cycle entry and state-specific fate outputs of tectal radial glia after injury" for consideration by *eLife*. Your article has been reviewed by three peer reviewers, and the evaluation has been overseen by Marianne Bronner as the Senior and Reviewing Editor.

The reviewers have discussed the reviews with one another and the Reviewing Editor has drafted this decision to help you prepare a revised submission.

The manuscript has been much improved but there remain some concerns about the grammar, use of arrows in figures, interpretation and novelty, plus one suggested experiment that needs repeating. I ask you to address these concerns in a revised version of the manuscript and refer you to the reviews below for details.

Reviewer #1:

1) This version of the manuscript is much improved and more readable; however, there are still many grammatical errors throughout (e.g. "low vertebrates") that should be addressed prior to publication. There are many professional editing services available.

2) I still do not see the mosaic pattern of expression of the mitotic cells after injury. For example, Figure 4F2 shows a large cluster of PCNA^+^ cells. Same for Figure 2D. If only about 25% of the cells in an area re-enter the cell cycle, random chance would predict that sometimes they are adjacent and sometimes they are not. This is not the definition of a mosaic. A mosaic has a regular, alternating pattern. If the authors were to make a flatmount of the tectum and image in the plane orthogonal to their sectioning at the ventricular surface they could show regular spacing if this exists, but the data are presented in a way where it is not possible to determine rigorously whether there is a regular mosaic or just randomly distributed cells that enter the cell cycle within a population. I am not sure it really matters to their argument in any case, but it is a good idea to use these terms with precision. If they cannot show this more directly, they should just delete the terms like "mosaic" or "salt and pepper" from the manuscript.

3) In the previous version the authors used an excessive number of arrows on some of the figures to the extent that this obscured the image (e.g. Figure 5L and 5M. They have addressed this issue by making these arrows open instead of white, but not reduced their numbers. This does not help much, and they should remove most of these from the panels. If they did so, it would be apparent that in the example section they show in Figure 5L1, the number of PCNA^+^ cells in the TS increases in the NICD over-expression. This raises the issue as to how the TS and PGZ are defined. The DAPI does not seem to help resolve this.

Reviewer #2:

This paper is a new submission of a previous manuscript entitled "Notch-dependent stochastic cell-cycle entry of injury-reactivated tectal radial glia".

In this manuscript the authors investigate the cellular and molecular response to optic tectum injury in adult zebrafish. They show that upon stab injury radial glial cells (RGCs), which are otherwise quiescent, become highly proliferative. They found that injury activated RGCs enter the cell cycle in a mosaic manner which is dependent on the Notch/Delta pathway. Their data also suggest that in contrast to the zebrafish adult telencephalon, where activated RGCs will form new neurons, in the optic tectum the majority of the proliferative RGCs will give rise to new RGCs by gliogenesis.

In my opinion, in this new version the authors greatly improved the presentation, description and also the scientific content of their results. I am mostly happy with the data shown in this paper, although there are still several discrepancies between the findings reported in this work and the data and conclusions from Ueda et al., 2018. For example, the authors show here that post-injury inhibition of Notch results in an important increase in neurogenesis, while Ueda et al. observed a reduction. This difference is explained by a difference in the time window of Notch inhibition, which is not a very convincing explanation in my opinion. I would suggest to repeat the Notch inhibition using both time windows (i.e. those described in this paper and Ueda et al.)

My biggest concern is about the novelty of the findings of this manuscript, I am not really sure that the role of Notch described in this paper is really new and of highest priority to be published in a journal such as *eLife*.

In summary, the manuscript quality is improved but the scientific content and message of this work are still not of highest importance.

Reviewer #3:

I continue to feel that this a well-executed and documented study. The other reviewers raised questions about novelty and mechanistic depth, which I do acknowledge as legitimate concerns. However, as somewhat of an outsider to this field, I think this manuscript has considerable useful information, particularly with respect to the gene expression analysis performed by single cell RNA-seq.

---

## [Author Response]

[Editors’ note: the author responses to the first round of peer review follow.]

The following revisions are required and described in further detail in the full reviews below:• This paper needs to be rewritten.

Thanks for the comment. We have rewritten the entire paper and our native-speaker colleague also carefully edited the manuscript to ensure its clarity.

• New Introduction should be provided with more focus on fish data.

Thanks for the comment. We have rewritten the Introduction and the new Introduction is provided with more focus on fish data.

• The Results section, figures and figure legends need to be reorganized and, in some case, extended.

Thanks for the comment. We have reorganized and extended the Results section, figures and legends to make the paper more logical and rigorous.

• Stab wound Injuries need to be done at different geographical regions of the optic tectum.

Thanks for your comment. To answer this question, we completed two sets of new experiments.

First, we performed single-site injury on five different regions of the optic tectum. We found that the number of PCNA^+^ RG was similar across regions except for the medial-dorsal region (Figure 1—figure supplement 2F-P).

Second, we performed a two-sites injury on the same hemisphere of the optic tectum, the medial-dorsal and the central-dorsal region, to confirm this result (Figure 1—figure supplement 2Q-S). Consistently, significantly more PNCA^+^ RG were found in the central dorsal region than in the medial-dorsal region (Figure 1—figure supplement 2T).

• The capacity of the optic tectum to regenerate or not need to be investigated and discussed.

Thanks for your comment. We further investigated and discussed this question in our new submission.

RG in the optic tectum were able to generate newborn neurons, but the number was low (Figure 6C-H; 4 ± 1 cell; 3.3% ± 0.9% of total EdU^+^ cells; mean ± SEM, n = 8); 2. The stab wounds could not be well restored. We found hypertrophic processes of RG near the injury sites formed a glial scar-like structure which may block the restoration of the wounds (Figure 6—figure supplement 1C-I_3_). Altogether, our results indicated the regeneration capacity is not as strong as other brain regions in zebrafish, such as telencephalon where the wound could be restored perfectly (Kroehne et al., 2011).

• They need to show to which structure or tissue the newly generated RG cells or neurons (late treatment of notch) are giving rise.

Thanks for your comment. We further investigated and discussed this question in our new submission.

1) We found the newborn RG formed a glial bulge-like structure underneath the injury sites (Figure 6D), and this structure remained up to 300 dpi (Figure 6J-K_3_).

2) We investigated the long-term fate of the over-produced neurons in the Notch-inhibited (3-5 dpi) optic tectum. Interestingly, we found the number of EdU^+^/HuC/D^+^ newborn neurons in LY411575-treated fish decreased significantly at 25 dpi and became indistinguishable from the DMSO-treated fish (Figure 7Q, R and U). However, those remaining neurons could survive up to 86 dpi (Figure 7—figure supplement 1G-H). 76% of those long-lived neurons resided in the tectum opticum (the upper region of optic tectum) in 25-dpi LY411575-treated fish, which was similar with the control fish (Figure 7—figure supplement 1I-K).

Reviewer #1:[…] In principle, this is an interesting topic in a very important field of research.However, in my opinion, the majority of the conclusions drawn from the presented work are not novel.For example, the activation of RG cells upon injury and the role of notch in RG cell state and fate have been largely addressed in several earlier papers from the labs of Toshio Ohshima and Laure Bally-Cuif. I particularly invite the authors to refer to the publication by Ueda et al. from 2018, which deals with the role of notch signaling in regenerative neurogenesis in the adult zebrafish optic tectum (please cite this paper in your next submission).

Thanks for your comments. We have cited and discussed Ueda et al., 2018 and Chapouton et al., 2010in the new version. But our current study is the first one showing the role of Notch/Delta interaction in controlling the quantity of cell cycle entry of RG after injury.

Figures and figure legends are not well organized, which makes it difficult for the reader to follow and understand them, for example:- Figure 1A consists of 4 panels which are not labelled (the same is true for other figures).

Thanks for your comment. We have reorganized all figures. Also, we have labelled all of the panels to make it easier to read. We also reorganized our figures to ensure the clarity of the study.

- There are 2 panels labelled 1B but no panel 1C.

Thanks for your comment. Sorry for this mistake. We have corrected it in our new submission.

- In Figure 1G: where is the cell count for RG cells?

Thanks for your comment. We have counted and shown the quantifications of newborn cells and newborn RG in our new submission (Figure 6N).

- The figure legend of Figure 1 and in general all other figure legends in this work are very poor and cryptic. The authors need to better describe the figures. The reader needs to be guided through the results and the figures and should not have to guess what the authors want to show.

Thanks for your comments. We have extended the description of all experiments and the figure legends.

In Figure 2, at 5 dpi apparently there is an increase in GFP expression and a lot of proliferation close to the site of injury. What kind of cells are these? Please describe and investigate this issue further. At least discuss it in the Discussion section.

Thanks for your comments. We further investigated this question in the new submission. We think the GFP signal close to the injury sites were the hypertrophic processes of RG under the injury sites (Figure 2C and D). We have performed injury on the optic tectum of oligodendrocyte labelling line Tg(*olig2*:EGFP) and microglia/macrophage labeling line Tg(*mpeg1*:EGFP), and found some oligodendrocytes and microglia/macrophages were also induced to proliferate upon injury (Figure 3—figure supplement 1D-H_3_). We also found some proliferative cells who were neither oligodendrocyte nor microglia/macrophage, they might be some other cell types which required further investigation (Figure 3—figure supplement 1D-H_3_).

For some injury experiments the authors use 2 months old zebrafish. Why? To my knowledge, zebrafish are considered adults only after 3 months post fertilization.

Thanks for your question. We used young adult fish from 2 to 4 months in our experiments as the fish are sexually matured.

In their manuscript the authors suggest that injury of the zebrafish adult optic tectum leads to gliogenesis. This result is very interesting, but partially in contradiction with previous results reported by Shimizu et al., 2018, and Ueda et al., 2018, which showed that injury induces proliferation of RG cells and subsequent formation of new neurons. These contradictions need to be discussed by the authors.

Thanks for your comments. We have discussed this in our new submission. Shimizu et al., 2018 and Ueda et al., 2018 showed RG generated some neurons in the injured optic tectum. The average number of newborn neurons in their study (~ 5 BrdU^+^/HuC/D^+^ neurons) was largely similar to what we had (~ 4 EdU^+^/HuC/D^+^ neurons), while the number of total newborn cells in their study (~ 15 BrdU^+^ cells) is much smaller than ours (~ 90 EdU^+^ cells). Thus, the conclusion of high portion of newborn neurons from Ohshima’s lab is perhaps due to the underestimated number of total newborn cells. Our quantification is rigorous, that is, every second 14-microns serial cryosections of the injured region was analyzed.

Galant et al. showed in 2016 that new neurons are generated only from RG cells of the PGZ, close to the tectal proliferation zone (TPZ) of the optic tectum. This suggests that the region where the injury is inflicted can decide between gliogenesis or neurogenesis.In their approach the authors observed an increase in gliogenesis but fail to describe and discuss the consequences. Do they see regeneration in the optic tectum? Is there scar formation? These are very important points which also need to be addressed.

Thanks for your comments. We have investigated and discussed these questions in our new submission. Also see our last response to reviewer #2.

1) We did observe many newborn cells after stab injury. Although the majority of the newborn cells were glial cells, a few of them were neurons, which could survive up to 300 days post injury (dpi).

2) However, we failed to see the restoration of the stab wound caused by stab injury. Although RG did not migrate to the injury sites, we did see the hypertrophic processes of RG around and at the injury sites formed a scar-like structure that may block the restoration of the wounds (Figure 6— figure supplement 1C-I_3_).

3) We also labeled the newborn cells derive from the tectal proliferation zone (TPZ) by 6 days injections of EdU, and analyzed the results at 400 days. We found the newborn cells migrated towards the central region of the optic tectum as a cell column. Among the cell column, we found most of the EdU^+^ cells were neurons and only the deepest layer were glia (see Author response image 1). This is consistent with the findings reported by Galant et al.(Galant et al., 2016). The result indicated that cells in the TPZ were different from injury-reactivated RG in term of fate potentials. In the TPZ, progenitor cells continuously generate many newborn cells (many neurons and a few RG) to keep the growth of optic tectum, whereas RG only respond to injury and largely generate glial cells and few neurons.

**Author response image 1. respfig1:** Neurogenic potential of progenitor cells in TPZ of adult zebrafish. (**A**) Experimental time courses of long-term tracing of RG in TPZ (panel B-D3). The fish are injected with EdU for 6 consecutive days and analyzed at day 400. (B-D_3_) Representative images of EdU (red), BLBP (gray) and HuC/D (green) immunofluorescences showing newborn cells in a column migrate toward (red arrow) the central region of optic tectum. Most of the newborn cells are neurons, only the deepest layer of newborn cells become new RG (white arrows in D-D_3_). Yellow dashed lines indicate the boundary of tectal ventricle. Yellow dashed lines indicate the tectal ventricle boundary. RG, radial glia; TeO, tectal opticum; PGZ, periventricular gray zone; TS, torus semicircularis; Val, valvula cerebelli. Scale bars, 100 μm (**B**); 20 μm (F-D_3_).

Reviewer #2:In this study the authors report that the proliferation of the radial glia of the zebrafish tectum after stab injury is dependent on Notch. The authors use a combination of immunolabeling and single cell RNA-seq to demonstrate this relationship, and do functional tests with Notch inhibitors and NICD over-expression. Similar results have been obtained in other areas of the nervous system in fish (e.g. retina), and so this result is not really surprising. This is a largely descriptive study, but what is needed is a mechanistic understanding of the connections between Notch signaling and proliferation in the fish glia, and this is lacking.1) The focus on a "salt-and-pepper" is a bit odd. It looks more like clumps of labeled cells around the site of the stab injury. (e.g. Figure 4D).

Thanks for your comment. Salt-and-pepper is used to describe proliferative and non-proliferative RG distributed in a mosaic manner.

2) The use of PCNA is the primary label for proliferating cells, but it is important to note that this is not a reliable indicator of proliferating cells. Additional markers, such as EdU labeling are necessary to substantiate this.

Thanks for your comment. PCNA is a widely used antibody of proliferating cells. As suggested, we also performed BrdU labeling and showed that radial glia enter S phase (Figure 6—figure supplement 1A-C).

3) In most experiments, small numbers of fish were analyzed with relatively small numbers of cell counted. The results should be substantiated with larger numbers.

Thanks for your comment. As suggested, we have completed 3-10 animals for each experiment. We also used 9-18 (≥ 3 replicates for each experiment) animals in the all of the new experiments shown in (Figure 7H-7U).

4) There are many grammatical errors throughout the manuscript and the authors should have this edited for English.

Thanks for your comment. We have invited a native speaker to carefully edit our manuscript.

5) In some figures, the excessive number and location of arrows prevents one from actually seeing the result (Figure 4E).

Thanks for your comment. As suggested, we changed the solid arrows to empty arrows in some panels such as Figure 4F-F_3_ and Figure 5L-M_3_., thanks.

[Editors' note: the author responses to the re-review follow.]

The manuscript has been much improved but there remain some concerns about the grammar, use of arrows in figures, interpretation and novelty, plus one suggested experiment that needs repeating. I ask you to address these concerns in a revised version of the manuscript and refer you to the reviews below for details.Reviewer #1:1) This version of the manuscript is much improved and more readable; however, there are still many grammatical errors throughout (e.g. "low vertebrates") that should be addressed prior to publication. There are many professional editing services available.

Thanks for your comments. We have decided to use “teleost fish” to replace “low vertebrates” in the revised version. Also, the revised manuscript has been edited by professional editing service.

2) I still do not see the mosaic pattern of expression of the mitotic cells after injury. For example, Figure 4F2 shows a large cluster of PCNA^+^ cells. Same for Figure 2D. If only about 25% of the cells in an area re-enter the cell cycle, random chance would predict that sometimes they are adjacent and sometimes they are not. This is not the definition of a mosaic. A mosaic has a regular, alternating pattern. If the authors were to make a flatmount of the tectum and image in the plane orthogonal to their sectioning at the ventricular surface they could show regular spacing if this exists, but the data are presented in a way where it is not possible to determine rigorously whether there is a regular mosaic or just randomly distributed cells that enter the cell cycle within a population. I am not sure it really matters to their argument in any case, but it is a good idea to use these terms with precision. If they cannot show this more directly, they should just delete the terms like "mosaic" or "salt and pepper" from the manuscript.

Thanks for your comments. We delete the terms like “mosaic” and “salt and pepper”, and use “a subset of RG enter the cell cycle” instead in the revised version.

3) In the previous version the authors used an excessive number of arrows on some of the figures to the extent that this obscured the image (e.g. Figure 5L and 5M. They have addressed this issue by making these arrows open instead of white, but not reduced their numbers. This does not help much, and they should remove most of these from the panels. If they did so, it would be apparent that in the example section they show in Figure 5L1, the number of PCNA^+^ cells in the TS increases in the NICD over-expression. This raises the issue as to how the TS and PGZ are defined. The DAPI does not seem to help resolve this.

Thanks for your comments. We have updated all of the main figures and supplementary figures by reducing the number of arrows and reducing the size of arrows.

Torus semicircularis (TS) is the midbrain tissue underneath the PGZ of the optic tectum, and their boundary can be unambiguously defined by DAPI staining (Figure 1—figure supplement 1A-C_2_ and Figure 5—figure supplement 1E-G_3_). Stab injury can induce some cells to become proliferative in the TS underneath the injury site (close to the boundary of TS and PGZ) in some animals.

Reviewer #2:[…] In my opinion, in this new version the authors greatly improved the presentation, description and also the scientific content of their results. I am mostly happy with the data shown in this paper, although there are still several discrepancies between the findings reported in this work and the data and conclusions from Ueda et al., 2018. For example, the authors show here that post-injury inhibition of Notch results in an important increase in neurogenesis, while Ueda et al. observed a reduction. This difference is explained by a difference in the time window of Notch inhibition, which is not a very convincing explanation in my opinion. I would suggest to repeat the Notch inhibition using both time windows (i.e. those described in this paper and Ueda et al.)

Thanks for the suggestion.

*Ueda et al.* recently showed that post-injury Notch inhibition resulted in a significant reduction of neurogenesis, which is in contrast to our findings as our results showed that post-injury Notch inhibition significantly promoted the neurogenesis. The reason for the observed difference is related to the time window of EdU or BrdU treatment was different. Ueda and colleagues treated the injured fish with BrdU during 2-3 dpi. Instead, we treated the injured fish with EdU during 4-6 dpi. This means the RG analyzed from the two studies were different.

Our analysis (Figure 2 and Figure 3) revealed that there were three different states of RG in the injured optic tectum, including dormant RG, reactive RG, and proliferative RG. In *Ueda et al.* study, BrdU treatment during 2-3 dpi is most likely to label proliferative RG as a result of the cell-cycle entry of dormant RG by stab injury, whereas EdU treatment during 4-6 dpi in our analysis is most likely to label proliferative RG as a result of the cell-cycle entry of injury-induced reactive RG by Notch inhibition instead.

Thus, our study and Ueda’s study indicate that the initial cell state influences the fate control of tectal RGs in response to either stab injury or Notch inhibition.

In the meanwhile, as suggested, we have repeated the experiments, and each group has four replicates (see Author response image 2). The new result showed when we labeled the injury-induced proliferative RG by injecting EdU during 1-3 dpi, 4-days’ Notch inhibition (4-7 dpi, Author response image 2) resulted in a significant reduction in the number of newborn neurons, which is consistent with Ueda and colleagues’ findings (Author response image 2). We further analyzed the proportions of newborn cells, which did not change significantly (Author response image 2).

**Author response image 2. respfig2:** 4-days’ Notch inhibition results in a significant decrease in the number of newborn neurons after injury. (**A** and **B**) Schematics of the experimental procedure for Notch inhibition experiments shown in (C-F). After the injury, fish are treated with LY411575 during either 4-5 dpi (**A**) or 4-7 dpi (**B**), and are injected with EdU for 3 consecutive days during 1-3 dpi. Control fish are treated with DMSO. All the fish are sacrificed and analyzed at 7 dpi. (C-I) Representative images of HuC/D (green), BLBP (blue) and EdU (red) immunofluorescences of the 7-dpi optic tecta after the treatments in (A and B). In both LY411575-treated and DMSO-treated optic tecta, only few newborn neuron (white arrowheads) is observed. (G-I) Quantifications of EdU^+^ newborn cells, EdU^+^/BLBP^+^ newborn RG and EdU^+^/HuC/D^+^ newborn neurons in (C-F). Neither 2-days’ nor 4-days’ Notch inhibition changes the number of newborn cells or newborn RG. 2-days’ Notch inhibition results in a decrease tendency of neuron production, whereas 4-days’ Notch inhibition significantly reduced neuron production (4 replicates for each group; mean ± SEM, ns, not significant, *p < 0.05; Wilcoxon test). (**M**) Ratios of EdU^+^/HuC/D^+^ newborn neurons to EdU^+^ newborn cells in (C-F). Neither 4-days’ nor 2-days’ Notch inhibition significantly changes the proportion of newborn neurons. (4 replicates for each group; mean ± SEM; ns, not significant; Wilcoxon test). The numbers above the bars indicate the animals used. Yellow dashed lines represent the tectal ventricle boundary. RG, radial glia; TeO, tectum opticum; PGZ, periventricular gray zone; TS, torus semicircularis. Scale bars, 30 μm. See also Figure 7 in main text.

My biggest concern is about the novelty of the findings of this manuscript, I am not really sure that the role of Notch described in this paper is really new and of highest priority to be published in a journal such as eLife.

Thanks for raising the concern. In our view, the novelties of our findings are listed below:

1) Sequential injuries of the same site lead to a distinct but overlapping population of tectal radial glia entering the cell cycle, suggesting the stochastic cell-cycle entry.

2) Post-injury Notch inhibition of non-proliferative reactive RG results in increased neurogenesis, suggesting the involvement of cell state in the fate control of injury-reactivated RG.

3) Single-cell RNA-sequencing data of tectal RG after the injury provides an enriched resource for future.